# Essential role of NONO-HOXA1-Wnt axis in cardiomyocyte differentiation

Zhiyu Feng[1,8], Yuan Gao[1,8], Han Gao[1,8], Siyu Sun[1,8], Weilan Na[1,8], Xianghui Huang[2], Shuolin Li[1], Chaozhong Tan[1], Shaojie Min[1], Yuquan Lu[1], Quannan Zhuang[1], Siyi Lin[1], Xiaojing Ma[1], Ying Liu[3], Weinian Shou[3], Mei Wang[4], Jing Wang[4], Zhongkai Gu[5], Wei Sheng[1,2,6] ✉, Feizhen Wu[7] ✉ & Guoying Huang[1,2,6] ✉

NONO is recognized as a critical molecular scaffold involved in both transcriptional and posttranscriptional regulation. Mutations in *NONO* are frequently linked to congenital heart diseases (CHDs) in humans. However, the mechanisms by which NONO regulates cardiac development remain elusive. Here, we identified NONO as a pivotal dual-function regulator of cardiomyocyte differentiation in human induced pluripotent stem cells (hiPSCs). NONO deficiency in hiPSCs results in a distinct defect in early cardiomyocyte differentiation. Mechanistically, NONO interacts with HOXA1 and regulates the dynamic expression of key genes during early cardiomyocyte differentiation. ChIP-seq analysis reveals that NONO loss reduces HOXA1 occupancy at target genes, compromising its transcriptional regulation. Additionally, NONO and HOXA1 cooperatively activate the Wnt signaling. Taken together, these findings establish the NONO-HOXA1-Wnt axis as a key molecular mechanism in cardiomyocyte differentiation and provide insights into the etiology of CHDs associated with *NONO* mutations.

Congenital heart disease (CHD) is the most common birth defect, affecting ~1% of live births and leading to significant morbidity and mortality[1,2]. Heart development requires the precise regulation of gene expression, primarily via cardiac transcription networks governing early cellular differentiation and later cellular maturation[3]. These networks cooperate to finely tune the regulatory machinery for gene expression, which is essential for the proper spatiotemporal control of cardiac morphogenesis and provides the foundation for proper cardiac physiology[4,5].

The non-POU domain-containing octamer-binding protein (NONO) is characterized as an important molecular chaperone that is best known for its association with molecular machinery involved in

both transcriptional and posttranscriptional regulation. NONO is widely expressed and is encoded by an X-linked gene that contains two RNA recognition motifs, a NonA/paraspeckle domain, and coiled-coil and intrinsically disordered regions at the N- and C-termini[6,7]. This unique structure enables NONO to bind both DNA and RNA, facilitating its roles in transcription regulation, mRNA splicing, and DNA double-strand break repair[8–12]. Our previous work demonstrated that NONO recruits 10–11 translocation 1 (TET1) to specific genomic loci in mouse embryonic stem cells (mESCs), thereby modulating 5-hydroxymethylcytosine (5hmC) levels and playing a critical role in neuronal differentiation[13]. More importantly, in the clinic, dysfunctional variants of *NONO* have been identified in a

[1]Shanghai Key Laboratory of Birth Defects, Pediatric Heart Center, Children's Hospital of Fudan University, Shanghai, China. [2]Fujian Key Laboratory of Neonatal Diseases, Children's Hospital of Fudan University at Xiamen (Xiamen Children's Hospital), Fujian, China. [3]Herman B Wells Center for Pediatric Research, Department of Pediatrics, Indiana University School of Medicine, Indianapolis, IN, USA. [4]Shanghai General Hospital, Shanghai Jiao Tong University School of Medicine, Shanghai, China. [5]Institutes of Biomedical Sciences of Fudan University, Shanghai, China. [6]Research Unit of Early Intervention of Genetically Related Childhood Cardiovascular Diseases (2018RU002), Chinese Academy of Medical Sciences, Shanghai, China. [7]Intelligent Medicine Institute of Fudan University, Shanghai, China. [8]These authors contributed equally: Zhiyu Feng, Yuan Gao, Han Gao, Siyu Sun, Weilan Na. ✉e-mail: sheng4616@126.com; wufz@fudan.edu.cn; gyhuang@shmu.edu.cn

broad spectrum of CHDs and cardiomyopathies, including atrial septal defect (ASD), ventricular septal defect (VSD), pulmonary stenosis (PS), right aortic arch (RAA), Ebstein anomaly, and left ventricular noncompaction (LVNC)[14–17], particularly in male patients. The birth rate of Nono-KO mice is significantly lower than that of wild-type littermates, and for those that survive, Nono-KO mice exhibit impaired diastolic cardiac function, leading to heart failure and early lethality[18,19]. Despite these previous observations, the biological role and underlying molecular mechanisms by which NONO regulates cardiogenesis and cardiomyocyte (CM) differentiation are poorly understood.

In this study, we tested the hypothesis that NONO is critical for heart development by employing hiPSC-directed CM differentiation as a model system. Our findings revealed that NONO is essential for proper CM differentiation. We demonstrated that the DNA-binding domain of NONO directly interacts with the homeobox domain of homeobox A1 (HOXA1), a master transcription factor for establishing the anterior–posterior axis during embryonic development[20]. This NONO-HOXA1 interaction facilitates the association of HOXA1 with chromatin and promotes the expression of genes in the cardiac and Wnt pathways, such as *MESP1, PDGFRA, MIXL1,* and *WNT5A*. NONO deficiency impairs this interaction, leading to the disrupted regulation of the Wnt signaling pathway and abnormal CM differentiation. Overall, our study reveals the NONO-HOXA1-Wnt axis as a critical mechanism driving CM differentiation and provides insights into the potential pathogenesis of CHDs and cardiomyopathies caused by *NONO* dysfunctional variants in humans.

## Results

### *NONO* depletion impairs CM differentiation

Previous studies have established that the mutations in *NONO* are frequently associated with various types of CHDs[17]. To elucidate the mechanisms underlying how *NONO* mutations cause CHDs, we employed an episomal vector-based CRISPR/Cas9 (Epi-CRISPR) gene-editing approach to generate *NONO* knockout human induced pluripotent stem cells (hiPSCs), and two NONO-KO hiPSC lines were obtained (Supplementary Fig. 1A, B). Karyotype analysis of WT-hiPSCs and NONO-KO clones (3 and 11) confirmed stable, normal chromosomal integrity (Supplementary Fig. 1C). As all NONO-KO hiPSCs presented similar phenotypes, NONO-KO clone-3 cells were used in subsequent analyses. These NONO-KO hiPSCs retained normal pluripotency (Supplementary Fig. 1D, E). We then differentiated these cells into CMs following previously established protocols (Supplementary Fig. 2A)[21]. First, we analyzed the temporal expression patterns of key cardiac differentiation factors, including *OCT4, T, EOMES, MESP1, ISL1, GATA4, PDGFRA, NKX2-5, MYH6, MYH7,* and *TNNT2* within our in vitro cardiac differentiation system using qRT–PCR (Supplementary Fig. 2B). Day 15 scRNA-seq revealed that TNNT2+ CMs exhibited high MYL7 (73.9%) and low MYL2 (0.4%) expression (Supplementary Fig. 2C), consistent with an immature, atrial-like phenotype as previously reported[21,22]. Expression analysis of cardiac lineage markers from Day 2 to Day 7 revealed that NONO-KO hiPSCs exhibited substantially reduced induction of *T, EOMES, MIXL1, MESP1, PDGFRA, GATA4, NKX2-5,* and *MYOCD* compared with WT cells, highlighting the essential role of NONO in early CM differentiation (Supplementary Fig. 2D). Flow cytometry demonstrated reduced proportions of PDGFRA+ (Day 3), GATA4+ (Day 5), and NKX2-5+ (Day 6) cells in NONO-KO cultures compared with WT (67.0% vs. 83.5%; 18.3% vs. 35.0%; 41.2% vs. 77.2%) (Supplementary Fig. 2E–J). At Day 15, the percentage of cTNT+ CMs was diminished in NONO-KO cells (11.6%) relative to WT (70.9%) (Fig. 1A), about a 59.3% reduction in CM differentiation. Consistent with the notion, qRT–PCR analysis further revealed largely reduced expression levels of major myofibril genes, such as *ACTN2, TNNT2, MYL2, MYL7, MYH6, MYH7, TITIN, TNNI1,* and *TNNI3,* in NONO-KO CMs (Fig. 1B).

Immunofluorescence analysis revealed that ~80% of NONO-KO CMs on Day 15 presented disorganized sarcomere structures, in contrast with the well-organized sarcomeres observed in WT CMs (Fig. 1C, D). Transmission electron microscopy (TEM) confirmed those findings, revealing a lack of Z-disc structures in NONO-KO CMs, whereas WT CMs exhibited well-defined sarcomeres (Fig. 1E). Compared with WT CMs, NONO-KO CMs presented irregular contraction patterns and a reduced beating rate (Fig. 1F). To investigate these irregularities, we assessed spontaneous $Ca^{2+}$ transients, a critical physiological marker closely linked to CM-beating rates and myocardial functionality. NONO-KO CMs displayed irregular calcium transients (Fig. 1G right). We developed an R-based script for analyzing and visualizing calcium transient signals (see Methods for details) (Fig. 1H), which clearly highlighted aberrant patterns in NONO-KO CMs. These abnormalities were characterized by a significantly prolonged decay time, an increased time to reach peak and an extended peak-to-peak duration (Fig. 1I). Consistent with these observations, qRT–PCR analysis revealed reduced expression levels of key calcium-handling genes, including *RYR2* (calcium release), *SERCA2a* (calcium reuptake), and *CACNA1C* (calcium influx)[23] (Fig. 1J).

Differentiation of hPSCs into CMs generates cultures containing multiple lineage intermediates alongside the predominant CM population[22]. To examine how the loss of NONO affects these developmental trajectories, we performed time-course single-cell RNA-seq on WT and NONO-KO cells collected at Days 2, 5, and 15 of differentiation. UMAP embedding resolved major developmental populations—including mesendoderm, cardiac progenitors, CMs, endoderm, epithelial, endothelial, and neural progenitor cells—which were consistently identified across samples based on canonical marker expression (Supplementary Fig. 3A, B). Cells from all replicates were well integrated, indicating minimal batch effects (Supplementary Fig. 3C). Monocle2[24] trajectory analysis further demonstrated a continuous differentiation path from mesendoderm to cardiac progenitors and ultimately CMs in WT cells. However, NONO-KO cells showed hindered progression along the cardiac trajectory, with a subset retaining mesendoderm-like features at Day 5. Cardiac progenitor markers such as *ISL1, GATA4, HAND1,* and *TBX20* exhibited attenuated induction, accompanied by inappropriate persistence of *POU5F1*, resulting in a markedly lower CM yield at Day 15 (Fig. 1K, L, Supplementary Fig. 3D). CytoTRACE[25] scores showed that NONO-KO cells remained less differentiated than WT cells at both CPC and CM stages (Fig. 1M). Collectively, our results highlight that NONO is essential for CM differentiation. NONO deletion reduces differentiation efficiency, which is characterized by significantly fewer cTNT-positive cells, and disrupts CM function, as demonstrated by disorganized sarcomeres and impaired calcium handling in the low percentage of remaining differentiated CMs. Single-cell analyses show that NONO is required for the timely exit from the mesendoderm state. In the absence of NONO, Day 5 progenitors retain mesendoderm signatures and exhibit attenuated expression of key cardiac regulators, thereby failing to acquire the transcriptional competence necessary for efficient CM production.

### Reactivation of NONO expression in NONO-KO hiPSCs rescues CM differentiation

To determine whether reactivating NONO can rescue CM differentiation defects in NONO-KO hiPSC, we utilized a PiggyBac (PB) transposon-based doxycycline-inducible system to reactivate NONO expression in NONO-KO hiPSC lines[26]. To mimic the natural expression pattern of *NONO* during differentiation (Supplementary Fig. 4A), doxycycline was administered from Days 1–2 of differentiation (Fig. 2A). The generated NONO-RE (NONO-KO^NONO OE with Dox) clones exhibited inducible NONO expression upon doxycycline treatment. Compared with wild-type cells, NONO-RE cells showed only a partial rescue at Day 2, as confirmed by Western blot

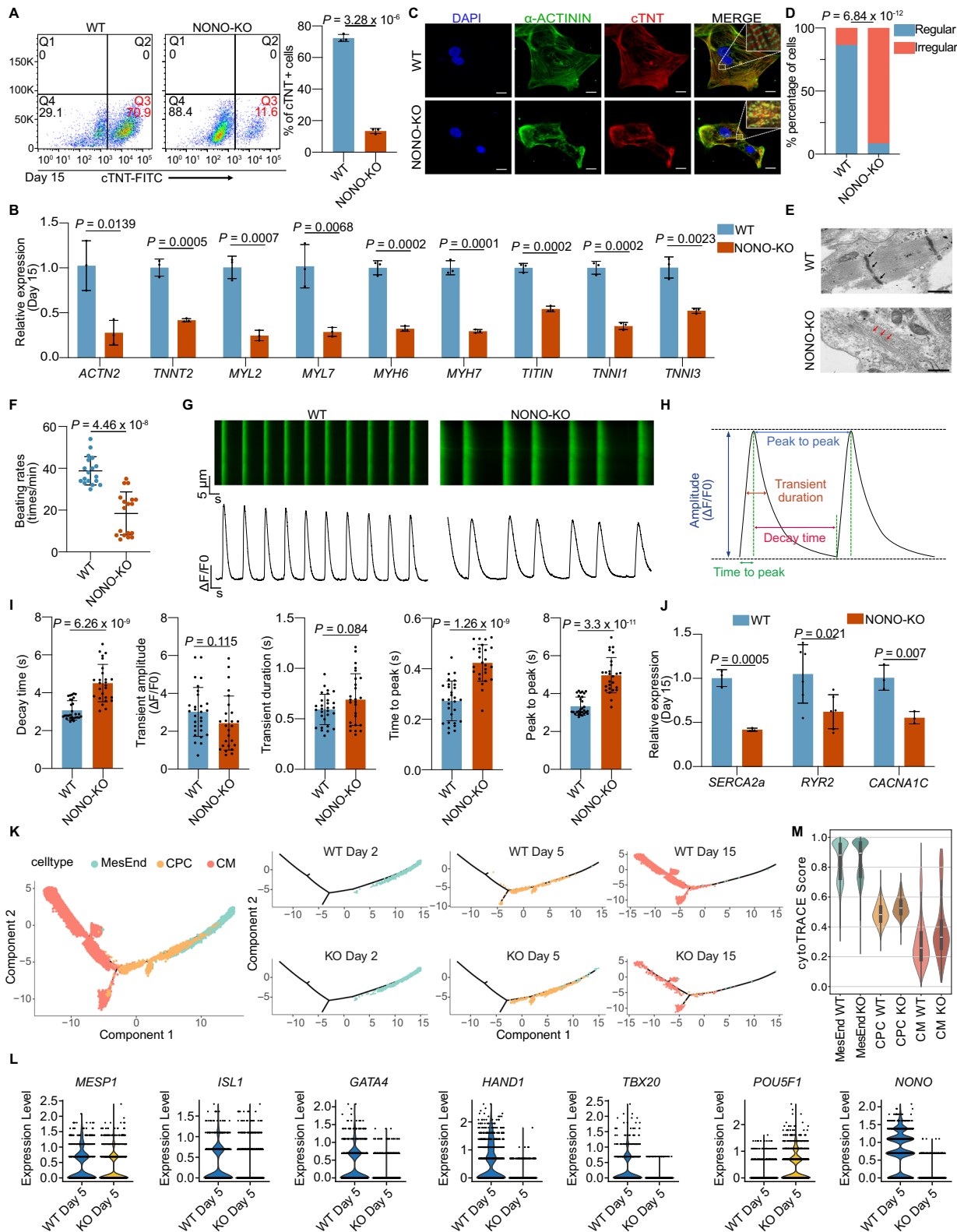

analysis (Supplementary Fig. 4B, C). To assess whether NONO reactivation could restore early cardiac specification in NONO-KO cells, we examined the expression of lineage markers from Day 2 to Day 7 by qRT-PCR. NONO reactivation reinstated the induction of *T*, *EOMES*, *MIXL1*, *MESP1*, *PDGFRA*, *GATA4*, *NKX2-5*, and *MYOCD* (Fig. 2B, C). By Day 15, flow cytometry analysis of differentiated cells revealed a significant improvement in the proportion of cTNT-positive cells

among NONO-RE (29.5%) compared with that observed among NONO-KO^NONO OE without Dox (4.07%), ~25.43% enhancement in CM differentiation (Fig. 2D, E). Immunofluorescence analysis revealed a significant improvement in sarcomere formation in NONO-RE CMs, and TEM revealed the restoration of Z-line structures (Fig. 2F–H). qRT–PCR analysis of the expression levels of major sarcomeric genes also demonstrated marked recovery (Fig. 2I).

**Fig. 1 | *NONO* depletion disrupts cardiomyocyte differentiation and function.**
**A** Flow cytometry analysis and quantification of cTNT expression in hiPSC-derived cardiomyocytes at Day 15 of differentiation. *n* = 3. **B** qRT-PCR analysis of key sarcomeric gene expression in Day 15 cardiomyocytes. *n* = 3. **C, D** Immunostaining of α-ACTININ (green) and cTNT (red) in Day 15 cardiomyocytes (**C**). Scale bar, 15 μm. Sarcomere organization is quantified in (**D**) (*n* = 4; 37 cells/group analyzed). **E** Representative transmission electron microscopy images of Day 15 cardiomyocytes. Black arrows indicate organized Z-disks; red arrows indicate disrupted Z-disks. Scale bar, 500 nm. *n* = 3. **F** Beating rates of Day 15 cardiomyocytes (*n* = 3; 18 cells/group analyzed). **G** Representative images of spontaneous calcium transients of WT and NONO-KO differentiated cardiomyocytes at Day 15. **H** Schematic diagram of spontaneous calcium transient parameters. **I** Quantification of calcium transient parameters in day 15 cardiomyocytes. (*n* = 4; WT, 29 cells; KO, 26 cells). **J** qRT-PCR of calcium-handling genes at Day 15 (SERCA2a/CACNA1C, *n* = 3; RYR2,

*n* = 6). **K** Reconstruction of cell differentiation trajectory for WT and NONO-KO cells at Days 2, 5, and 15, showing Mesendoderm (MesEnd), Cardiac progenitor cell (CPC), and Cardiomyocyte (CM) lineages. **L** Violin plots showing expression levels of key cardiac lineage regulators and *NONO* within CPC populations at Day 5. **M** Violin plots show CytoTRACE scores across the indicated cell types and genotypes, depicting the min–max range and density. Embedded box plots indicate the median (center line), the interquartile range (Q1–Q3), and whiskers extending to the minimum and maximum values. Data are pooled from three independent biological replicates. Data are presented as mean values ± SD (**A, B, F, I, J**). Data in (**M**) are presented as median and min–max range. *P* values were calculated using a two-tailed unpaired Student's *t* test (**A, B, F, I, J**) or Fisher's exact test (**D**). *P* < 0.05 was considered significant. *n* represents independent biological replicates. Source data are provided as a Source Data file.

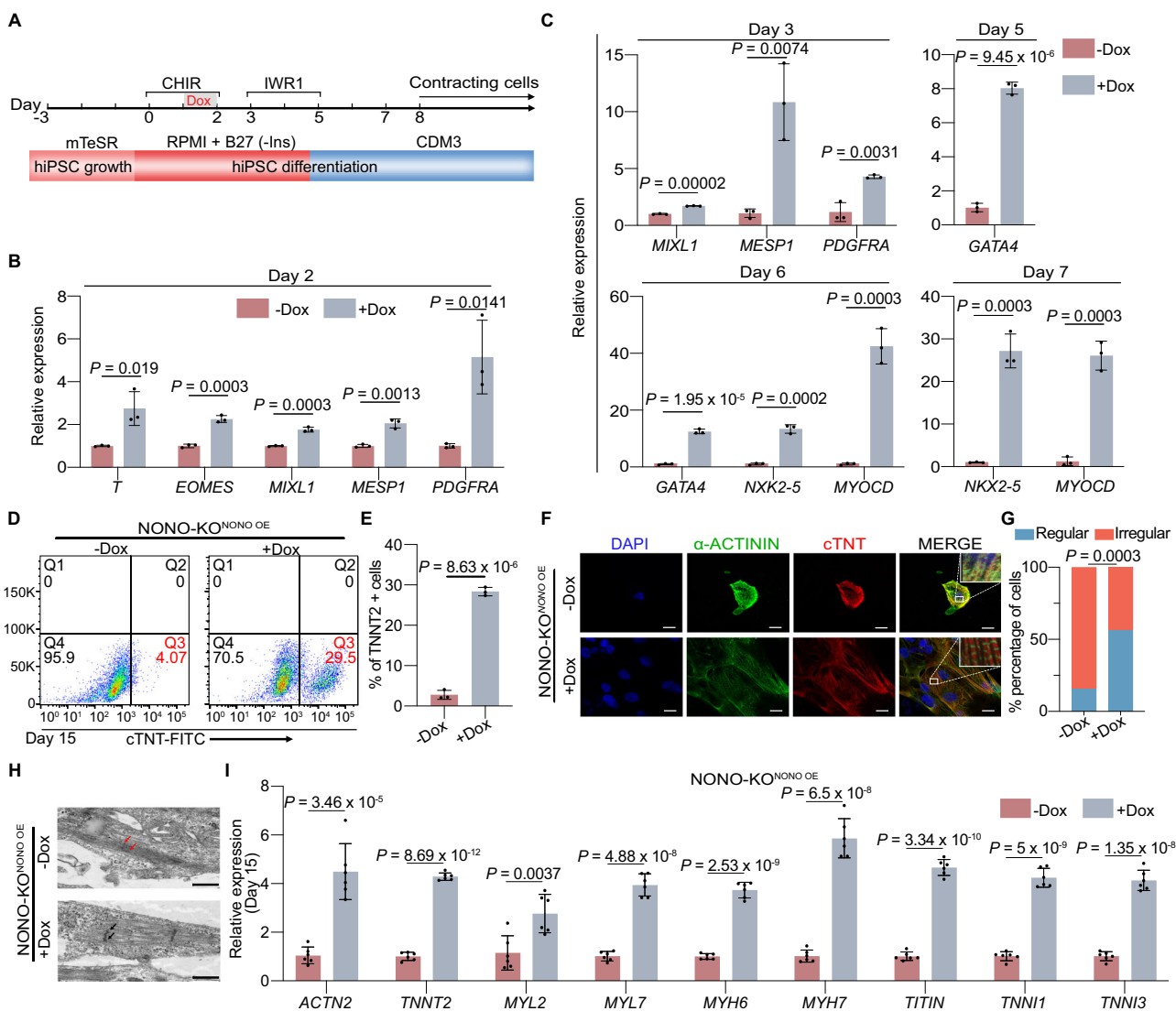

**Fig. 2 | Reactivation of NONO in NONO-KO hiPSC during the 1–2 days of differentiation rescues cardiomyocyte differentiation.** **A** Schematic illustration of the doxycycline (Dox)-inducible rescue strategy. NONO expression was induced in NONO-KO cells by Dox administration from Day 1–2 of cardiomyocyte differentiation. **B, C** qRT-PCR analysis of cardiac differentiation markers in NONO-KO^NONO OE cells from Days 2–7. Genes analyzed include *T, EOMES, MIXL1, MESP1*, and *PDGFRA* (**B**); and *MIXL1, MESP1, GATA4, NKX2-5*, and *MYOCD* (**C**). *n* = 3. **D, E** Flow cytometry analysis (**D**) and quantification (**E**) of cTNT⁺ cardiomyocytes at Day 15. *n* = 3. **F, G** Immunostaining (**F**) and quantification (**G**) of sarcomere

organization (α-ACTININ, green; cTNT, red) in Day 15 NONO-KO^NONO OE cardiomyocytes. Scale bar, 15 μm. (*n* = 4; with Dox, 39 cells; without Dox, 38 cells). **H** Representative TEM images of NONO-KO^NONO OE differentiated cardiomyocytes at Day 15. Black arrows indicate organized Z-disks; red arrows indicate disrupted Z-disks. Scale bar, 500 nm. *n* = 4. **I** qRT-PCR analysis of selected sarcomeric genes in Day 15 NONO-KO^NONO OE cardiomyocytes. *n* = 6. Data are presented as mean values ± SD. *P* values were calculated using a two-tailed unpaired Student's *t* test, except for (**G**) (Fisher's exact test). *P* < 0.05 was considered significant. *n* represents independent biological replicates. Source data are provided as a Source Data file.

To confirm those observations, we performed bulk RNA-sequencing using WT, NONO-KO and NONO-RE CMs on Day 15 of differentiation. A 9-square plot was used to determine potential genes that are directly associated with NONO-mediated regulation (see Methods for details). This in silico analysis, in which genes were categorized by a log2(1.5-fold changes) cutoff, revealed that the reactivation of NONO expression restored the expression of a significant number of NONO-modulated genes (Groups G and C) (Supplementary Fig. 4D), including essential cardiac genes such as *ACTN2* and *TNNT2* (Supplementary Fig. 4E). Gene Ontology (GO) enrichment analysis of these NONO-modulated genes highlighted significant enrichment in biological processes related to muscle contraction and cardiac muscle development (Supplementary Fig. 4F). Corresponding heatmaps illustrated the expression distribution of these genes, emphasizing the role of NONO in modulating genes essential for CM differentiation and function (Supplementary Fig. 4G, H). These findings reinforce the notion that NONO plays a critical role in CM differentiation.

### Pathogenic *NONO* variants fail to rescue CM differentiation defects

To evaluate the functional impact of clinically identified *NONO* variants, we reconstituted NONO-KO hiPSCs with doxycycline-inducible wild-type NONO (NONO-KO^WT-NONO OE) or four patient-specific truncation mutants (p.Arg73*, p.Arg184*, p.Arg337*, and p.Arg365*). Immunofluorescence staining for α-ACTININ and cTNT at Day 15 showed that re-expression of WT-NONO restored well-organized sarcomeric structures. In contrast, cells expressing the mutant variants displayed highly disorganized myofibrils and remained comparable to uninduced NONO-KO controls (Supplementary Fig. 5A, B).

Consistent with these morphological observations, flow cytometry analysis showed that WT-NONO robustly rescued CM differentiation, increasing the cTNT⁺ population to ~33.5% compared to ~4.0% in the −DOX control. In contrast, none of the four mutants improved differentiation, as cTNT⁺ fractions remained low (8.8%–11.9%) (Supplementary Fig. 5C, D). qRT-PCR further confirmed that WT-NONO, but not the mutant variants, restored expression of *TNNT2*, *MYH6*, and *MYH7* (Supplementary Fig. 5E). Collectively, these findings confirm that these clinical truncating mutations result in a loss of NONO function, thereby impairing CM differentiation.

### NONO dynamically regulates the expression of genes essential for CM differentiation

To investigate the molecular mechanisms underlying the effects of *NONO* depletion and subsequent rescue during CM differentiation, we assessed *NONO* expression levels via qRT–PCR. The results revealed that *NONO* expression peaked on Day 2, suggesting a critical role in the early stages of differentiation (Supplementary Fig. 4A). On the basis of these findings, we further analyzed mRNA-seq data of WT, NONO-KO, and NONO-RE hiPSCs at Days 2, 5, 9, and 15 of differentiation. Based on the 9-square plot to categorize genes on the basis of log2(1.5-fold change) in expression from Day 2 to Day 15 in WT (x-axis) and NONO-KO (y-axis) cells, the genes in Group I (*n* = 332) were upregulated during differentiation in the WT cells but downregulated in the NONO-KO cells, suggesting that these genes, which are normally activated during CM differentiation, were suppressed by NONO loss. In contrast, Group A genes (*n* = 565) presented the opposite pattern, indicating that dysregulation was caused by *NONO* depletion (Fig. 3A).

We then compared the gene expression levels in NONO-RE (*x* axis) and NONO-KO (*y* axis) cells between Day 2 and Day 15 of differentiation (Fig. 3B and Supplementary Fig. 6A). Among the 332 genes in Group I (Fig. 3A), 60 genes (18%) in Group I′ (Fig. 3B) presented restored expression in NONO-RE cells. This suggests that the dynamic expression of these genes during differentiation is dependent on NONO. To validate this dynamic regulation, we analyzed the expression trajectories of Group I′ genes from Day 2 to Day 15. The results revealed that

the expression of these genes was consistently downregulated in NONO-KO cells but upregulated by Day 15 in NONO-RE cells, a finding that aligns closely with the expression patterns observed in WT cells (Fig. 3C). Gene Ontology biological process (GO-BP) enrichment analysis revealed that Group I′ genes were significantly associated with cardiac muscle cell contraction processes (Fig. 3D). Notably, the expression of key cardiac genes, such as *GATA4*, *RYR2*, *KCNE2*, and *HCN4*, was restored in NONO-RE cells (Fig. 3E).

We then employed a 9-square plot to determine potential genes directly associated with NONO-mediated regulation during the early differentiation stages (Days 2 and 5). This analysis revealed that the reactivation of *NONO* expression restored the expression of a significant number of NONO-modulated genes (Groups G and C), including essential cardiac mesodermal and Wnt canonical genes such as *PDGFRB*, *MESP1*, *HAND1*, *WNT3A*, and *WNT11* (Supplementary Fig. 6B, C). The same analysis of upregulated and downregulated genes measured at different time points during CM differentiation (Days 5 and 2) revealed significant enrichment in cardiac ventricle morphogenesis (Supplementary Fig. 6D–G), emphasizing the pivotal role of NONO in regulating genes crucial for CM function. These findings demonstrate that NONO dynamically modulates the expression of genes critical for CM development.

### NONO regulates transcriptional activity independently of its role in pre-mRNA splicing function

Previous studies have identified NONO as a key regulator of both transcription and alternative splicing during retinal development[27]. In our experimental system, NONO deletion reduced cTNT-positive cell numbers and disrupted sarcomere structure and calcium handling, leading to impaired function. To determine whether these phenotypes are driven by transcriptional or splicing defects, we analyzed differentially expressed genes (DEGs) and alternative splicing events (ASEs) between WT and NONO-KO cells during differentiation (Supplementary Fig. 7A, B). We then examined the overlap between DEGs (4267), ASEs genes (242), and NONO-targeted genes (2212) on Day 2 of differentiation. NONO-targeted genes identified through NONO ChIP-seq were intersected with NONO-KO RNA-seq data, yielding 556 transcriptionally regulated genes. Additionally, 62 alternatively spliced genes were identified by intersecting NONO-KO RNA-seq data with splicing gene datasets. Among these genes, 14 were dually regulated by NONO through transcription and alternative splicing (Fig. 4A). Notably, the 542 transcriptionally regulated genes were enriched in heart development pathways, whereas the 48 genes exclusive to alternative splicing showed no significant functional enrichment (Supplementary Fig. 7C–F). Our results suggest that NONO regulates transcriptional activity independently of its splicing function, with these two roles likely influencing distinct cellular processes during early differentiation.

### NONO directly interacts with HOXA1

To investigate how NONO regulates the expression of genes involved in CM development, we examined its potential interactions with transcription factors critical for cardiac differentiation. Transcription factor networks play essential roles in the precise regulation of gene expression during cardiac development[4,28]. Previous studies have demonstrated that NONO participates in neuronal differentiation and hormone regulation as part of a complex[13,29]. We performed immunoprecipitation–mass spectrometry (IP–MS) analysis of NONO using Day 2 differentiated cells, corresponding to peak NONO mRNA expression. The results led to the identification of core components of the NONO complex, including PSPC1, NONO, and SFPQ. Notably, the transcription factor homeobox A1 (HOXA1), known to be involved in cardiac development, was also identified as part of the NONO complex (Supplementary Fig. 8A). *HOXA1* is an essential member of the Hox gene family and is pivotal for establishing the anterior–posterior axis

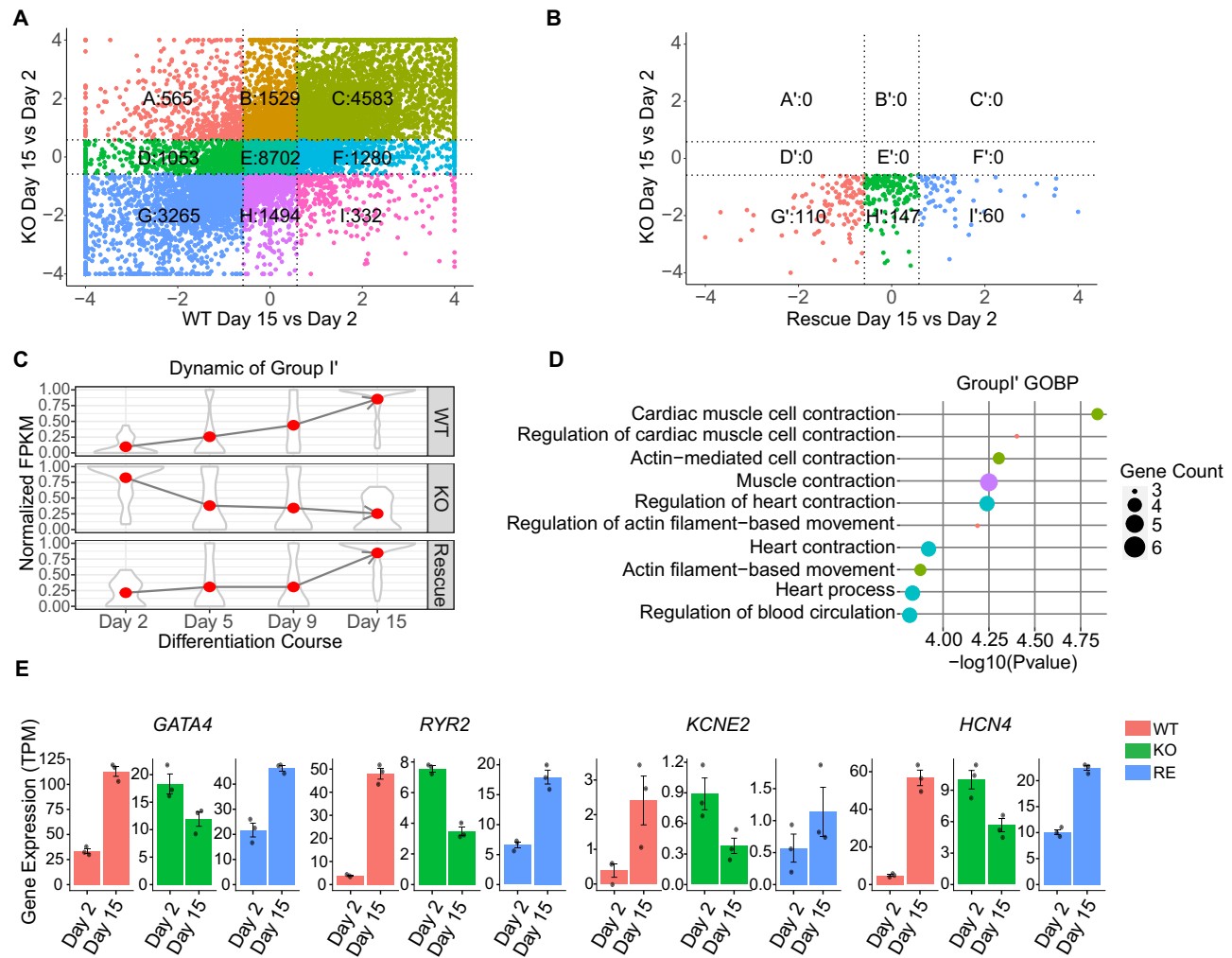

**Fig. 3 | Dynamic analysis of gene expression during cardiomyocyte differentiation. A** Comparison of log2 gene expression fold changes between day 2 and day 15 of cardiomyocyte differentiation in WT and NONO-KO cells reveals genes upregulated (Group I) or downregulated (Group A) in WT cells. The dotted lines indicate a log2 (1.5-fold change) cutoff. **B** Reanalysis of Group I genes comparing NONO-KO with NONO-RE (NONO-KO[NONO OE] with Dox) cells demonstrates that genes in Group I′ depend on *NONO* expression. **C** Box plots depict the dynamic expression patterns of Group I′ genes throughout the cardiomyocyte differentiation process from Day 2 to Day 15. The *y* axis displays normalized FPKM values, scaled from 0 to 1. These values represent the average FPKMs calculated from WT samples, KO samples, or combined NONO-RE samples. **D** GO biological process term enrichment analysis for Group I′ genes. **E** Bar plots display representative genes from Group I′: *GATA4*, *RYR2*, *KCNE2* and *HCN4*. *n* = 3. Data are presented as mean values ± SEM. *n* represents independent biological replicates.

during embryonic development[20]. Hox proteins typically function in conjunction with cofactors to precisely regulate gene expression[30], and *HOXA1* genetic mutations are associated with heart defects[31,32].

*HOXA1* genes are not expressed in ES cells but are activated during differentiation[33,34]. Consistently, both *NONO* and *HOXA1* exhibited synchronous increases in expression levels on Day 2, suggesting a coordinated expression pattern (Supplementary Fig. 4A, 8B). We validated the interaction between HOXA1 and NONO via co-immunoprecipitation (Co-IP) using HEK293T cells in which NONO-FLAG and HOXA1-HA were overexpressed. Co-IP results confirmed that HOXA1 coprecipitated with NONO and vice versa (Fig. 4B). To investigate the relevance of this interaction during differentiation, we generated a doxycycline-inducible HOXA1-HA overexpression hiPSC line (WT[HOXA1-HA OE]), in which HOXA1-HA was induced during Day 1 to Day 2, using the PiggyBac transposon system (Supplementary Fig. 8C). Co-IP of Day 2 differentiated cells with doxycycline-induced HOXA1-HA expression confirmed that endogenous NONO coprecipitated with HOXA1-HA, reinforcing the interaction between NONO and HOXA1 in the context of differentiation (Fig. 4C). Immunofluorescence analysis with anti-NONO and anti-HA antibodies demonstrated the nuclear

colocalization of NONO and HOXA1 in HEK293T cells (Fig. 4D) and Day 2 differentiated cells (Fig. 4E). Using a GST-pull-down assay, we confirmed that the GST-NONO fusion protein specifically binds to HOXA1-HA from HEK293T cell lysates (Fig. 4F and Supplementary Fig. 8D). These results indicate that NONO interacts with HOXA1 in the nucleus.

To identify the structural domains mediating the interaction between HOXA1 and NONO, we generated HOXA1 deletion mutants targeting three key regions: ΔPH (Poly-Histidine region deletion), ΔHP (Hexapeptide motif deletion), and ΔHD (Homeodomain deletion). In parallel, we constructed NONO deletion mutants lacking major functional domains: ΔRRM1 (RNA Recognition Motif 1 deletion), ΔRRM2 (RNA Recognition Motif 2 deletion), and ΔCoil (Coiled-coil domain deletion) (Fig. 4G). Co-IP analysis in HEK293T cells co-expressing NONO and HA-tagged HOXA1 fragments showed that ΔPH and ΔHP retained NONO binding, whereas deletion of the homeodomain (ΔHD) markedly impaired the interaction (Fig. 4H). Co-IP assays using FLAG-tagged NONO fragments further revealed that deletion of either RRM2 or the coiled-coil domain substantially reduced binding to HOXA1-HD (Fig. 4I). Together, these results demonstrate that the HOXA1 homeodomain and the NONO RRM2 and coiled-coil domains are essential

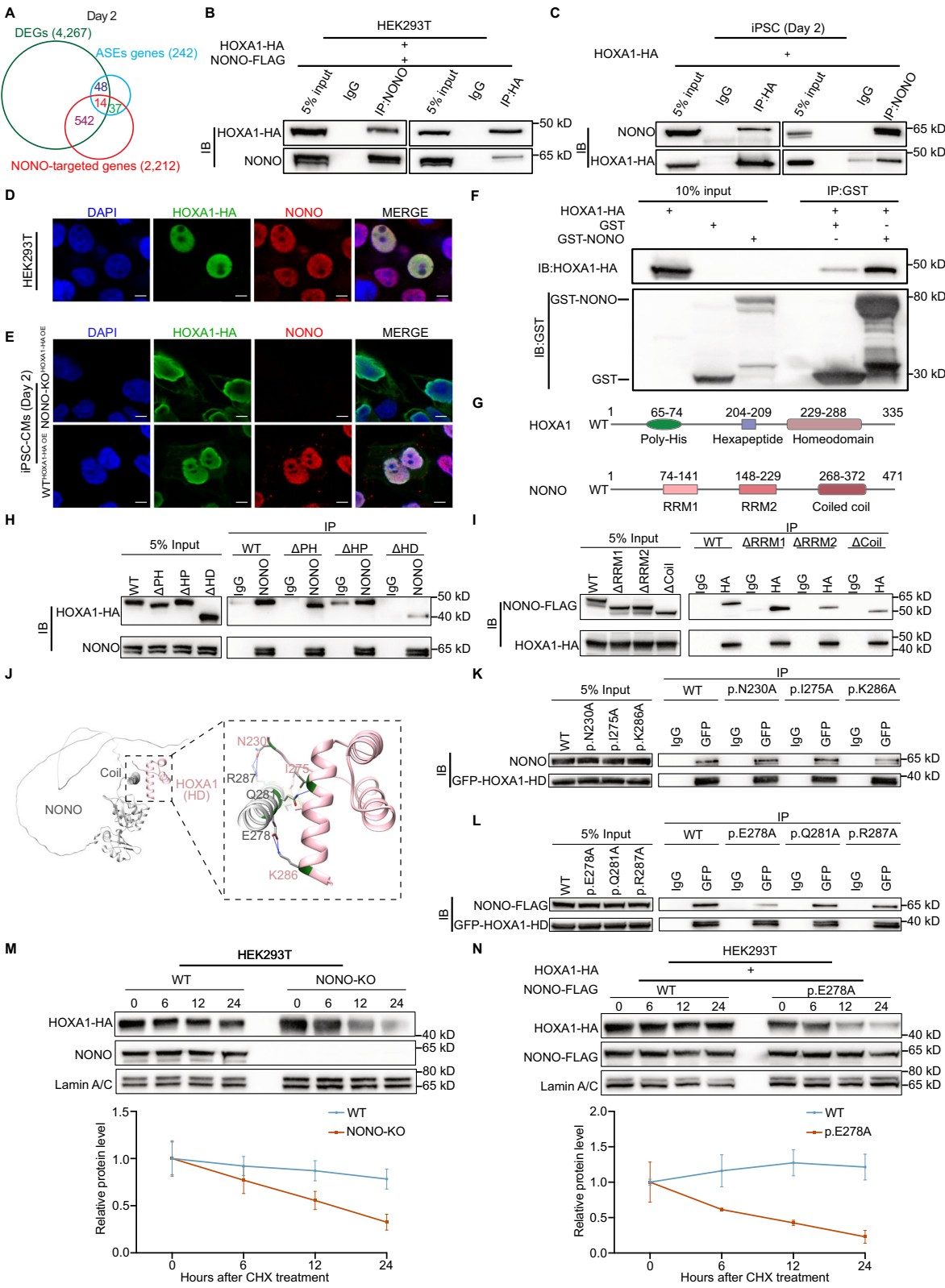

for mediating the NONO-HOXA1 interaction. To explore the structural basis of this interaction, we performed molecular modeling via the AlphaFold2 multimer[35]. The prediction indicated that residues N230, I275, and K286 within the HOXA1, together with E278, Q281, and R287 in NONO, form key hydrogen-bonding contacts that are critical for the NONO-HOXA1 interaction (Fig. 4J). To validate these predictions, we mutated these residues to alanine (HOXA1-HD: N230A, I275A, K286A; NONO: E278A, Q281A, R287A) to disrupt hydrogen bond formation.

Co-IP assays in HEK293T cells co-expressing NONO and GFP-tagged HOXA1-HD mutants showed that the K286A substitution markedly reduced NONO binding, indicating that K286 in HOXA1 is essential for the interaction (Fig. 4K). Similarly, Co-IP analyses using GFP-tagged HOXA1-HD and FLAG-tagged NONO mutants demonstrated that the E278A mutation substantially diminished binding to HOXA1-HD, establishing E278 in NONO as a critical determinant of the NONO-HOXA1 interaction (Fig. 4L). These findings demonstrate that the

**Fig. 4 | NONO directly interacts with and stabilizes HOXA1. A** Venn diagram overlapping alternatively spliced events (ASEs) genes, differentially expressed genes (DEGs), and NONO ChIP-Seq targets. **B, C** Co-immunoprecipitation (Co-IP) in HEK293T cells overexpressing NONO-FLAG and HOXA1-HA (**B**; $n = 1$), and in Day 2 hiPSC-derived differentiated cells with doxycycline-induced HOXA1-HA (**C**; $n = 3$), using anti-NONO and anti-HA antibodies for immunoprecipitation and detection. **D, E** Immunofluorescence staining showing nuclear colocalization of over-expressed HOXA1-HA and endogenous NONO in HEK293T cells (**D**) and in Day 2 differentiated cells (**E**). Specificity of the NONO antibody was validated in NONO-KO cells. Scale bar, 15 μm. $n = 3$. **F** GST pull-down assay using recombinant GST-NONO (from *E. coli*) and HOXA1-HA (from HEK293T lysates). $n = 2$. **G** Schematic representation of HOXA1 and NONO protein domains: Poly-His (PH), hexapeptide (HP), homeodomain (HD), RNA recognition motifs (RRM1, RRM2), and Coiled-coil (Coil) domain. **H, I** Co-IP domain mapping in HEK293T cells. Interaction of NONO with HOXA1-HA truncation mutants, showing diminished binding upon HOXA1-HD deletion (ΔHD) (**H**). Interaction of HOXA1-HA with NONO-FLAG truncation mutants, showing reduced binding upon NONO-RRM2 or Coil domain deletion (**I**). $n = 3$. **J** Structural model of the NONO-HOXA1 interface, highlighting key residues (HOXA1: N230, I275, K286; NONO: E278, Q281, R287). **K, L** Co-IP validation using point mutants in HEK293T cells. Results show disrupted interaction with the HOXA1-K286A (NONO) mutant (**K**) and the NONO-E278A (HOXA1) mutant (**L**) $n = 3$. **M, N** Cycloheximide (CHX) chase assay to assess HOXA1-HA stability. Quantification of HOXA1-HA in nuclear lysates of WT and NONO-KO cells (**M**). Quantification of HOXA1-HA in HEK293T cells co-expressing WT or E278A NONO-FLAG (**N**). Protein levels were normalized to LAMIN A/C and expressed relative to the 0 h time point. $n = 3$. Data are presented as mean values ± SD. *n* represents independent biological replicates. Source data are provided as a Source Data file.

interaction between HOXA1 and NONO requires key residues within both proteins: K286 in the HOXA1 and E278 within the NONO coiled-coil region. Together, these results establish a direct and residue-specific interaction between HOXA1 and NONO.

Given that NONO has been proposed to function as a protein chaperone, we examined whether it regulates HOXA1 protein stability. We generated NONO-KO HEK293T cells using the Epi-CRISPR system (Supplementary Fig. 1H). To examine whether NONO regulates HOXA1 protein stability, we overexpressed HOXA1-HA in WT and NONO-KO HEK293T cells and conducted cycloheximide (CHX) chase assays. We observed that HOXA1-HA decayed significantly faster in NONO-KO cells than in WT controls, indicating that NONO is required to maintain HOXA1 stability (Fig. 4M). To further elucidate the underlying mechanism, we co-expressed HOXA1-HA with either WT-NONO or the E278A mutant in WT HEK293T cells. In contrast to WT-NONO, the E278A variant failed to stabilize HOXA1 (Fig. 4N).

These results demonstrated that NONO binds to the HOXA1 homeodomain via its RRM2 and coiled-coil domains, with residues E278 of NONO and K286 of HOXA1 being critical for this interaction. Functionally, NONO acts as a chaperone to stabilize HOXA1 protein, a process strictly dependent on their physical association.

## NONO and HOXA1 co-occupy genes that are critical for pre-cardiac mesoderm differentiation

To explore the genomic landscape regulated by the NONO-HOXA1 axis during CM differentiation, we performed ChIP-seq in Day 2 differentiated cells. NONO profiling identified 4350 peaks, which were markedly enriched in promoter regions (53.55%) (Supplementary Fig. 8E) and targeted key early cardiac and mesoderm genes such as *EOMES*, *TBXT*, *DVL2*, and *MECOM*[36,37] (Fig. 5A). Gene Ontology (GO) analysis revealed the significant enrichment of NONO binding to genes involved in Wnt signaling, embryonic development regulation, and cardiac septum formation (Fig. 5B). Parallel HOXA1 ChIP-seq identified 26,143 peaks, predominantly located in intergenic and intronic regions (Fig. 5A, Supplementary Fig. 8F). GO analysis of HOXA1 target genes revealed enrichment in pathways related to Wnt signaling and cardiac development (Supplementary Fig. 8G). Bioinformatics analysis revealed 1169 overlapping peaks between the NONO and HOXA1 ChIP-seq datasets (Fig. 5C and Supplementary Fig. 8H), suggesting a shared chromatin association. GO analysis of these shared peaks revealed enrichment in processes such as the Wnt signaling pathway, gastrulation, and heart morphogenesis, suggesting a cooperative regulatory role for NONO and HOXA1 (Fig. 5D).

To further determine whether NONO and HOXA1 co-occupy the same genomic loci, we performed sequential ChIP (Re-ChIP) assays using a doxycycline-inducible HOXA1-HA line. HA-ChIP material was competitively eluted with HA peptide to preserve protein–DNA complexes before a second immunoprecipitation with anti-NONO. As visualized in the genome browser tracks, distinct Re-ChIP signals were successfully detected at the regulatory regions of key early cardiac and mesoderm drivers (Fig. 5A). These Re-ChIP peaks align precisely with the overlapping regions observed in the individual ChIP-seq profiles. Taken together, these findings suggest that NONO and HOXA1 co-occupy genomic regions associated with critical biological processes, including Wnt signaling, gastrulation, and heart morphogenesis, underscoring their cooperative role in CM differentiation.

## NONO is indispensable for the HOXA1-mediated regulation of precardiac mesoderm genes

Given that HOX transcription factors require cofactors for stable and specific DNA binding[38], we hypothesized that NONO and HOXA1 cooperate to regulate precardiac mesoderm genes. To test this hypothesis, we induced HOXA1-HA overexpression in WT$^{HOXA1-HA\ OE}$ and NONO-KO$^{HOXA1-HA\ OE}$ cells using Dox, and western blotting confirmed consistent overexpression efficiency (Supplementary Fig. 9A). HOXA1 ChIP-seq with an anti-HA antibody subsequently revealed 26,143 binding peaks in WT cells and 19,293 peaks in NONO-KO cells, with NONO-KO resulting in the loss of 12,483 HOXA1 binding peaks (Fig. 5E and Supplementary Fig. 9B, C). The loss of HOXA1 binding to target genes, including *MESP1*, *PDGFRA*, *HAS2* and *MIXL1* (Fig. 5F, G and Supplementary Fig. 9D, E), was closely linked to cardiac lineage development (Supplementary Fig. 9F). To determine whether the loss of these 12,483 HOXA1 binding peaks affects gene expression, we generated HOXA1-KO hiPSCs using the Epi-CRISPR system (Supplementary Fig. 1F, G), without an obvious effect on pluripotency (Supplementary Fig. 1D, E). We then performed a cross-analysis between the target genes associated with the 12,483 HOXA1 binding peaks lost upon NONO-KO and the DEGs identified in HOXA1-KO cells versus WT cells on Day 2 of differentiation. This analysis revealed 155 upregulated and 304 downregulated genes, indicating that these HOXA1-regulated genes are associated with HOXA1 promoter binding, which is dependent on NONO (Supplementary Fig. 9G). These DEGs were significantly enriched in cardiac development-related terms and the Wnt signaling pathway (Fig. 5H). Notably, 13 genes associated with cardiac development-related terms were also downregulated in NONO-KO cells (Supplementary Fig. 9H). qRT–PCR validation further confirmed the downregulation of these key precardiac mesoderm genes (*MESP1*, *PDGFRA*, *HAS2*, and *MIXL1*) in both HOXA1-KO and NONO-KO cells on Day 2 of differentiation (Fig. 5I and Supplementary Fig. 9I).

To elucidate the molecular mechanism underlying the impaired HOXA1 genomic occupancy in NONO-deficient cells, we examined the oligomeric states of these proteins. Using doxycycline-inducible systems, Co-IP assays first confirmed that NONO forms homodimers in differentiated cells (Fig. 5J). Crucially, we investigated whether NONO influences HOXA1 self-association. Co-IP analysis revealed that HOXA1 homodimer formation was markedly compromised in NONO-KO cells compared to WT controls (Fig. 5K, L). This suggests that NONO facilitates or stabilizes HOXA1 homodimerization, a conformational state often requisite for robust DNA binding.

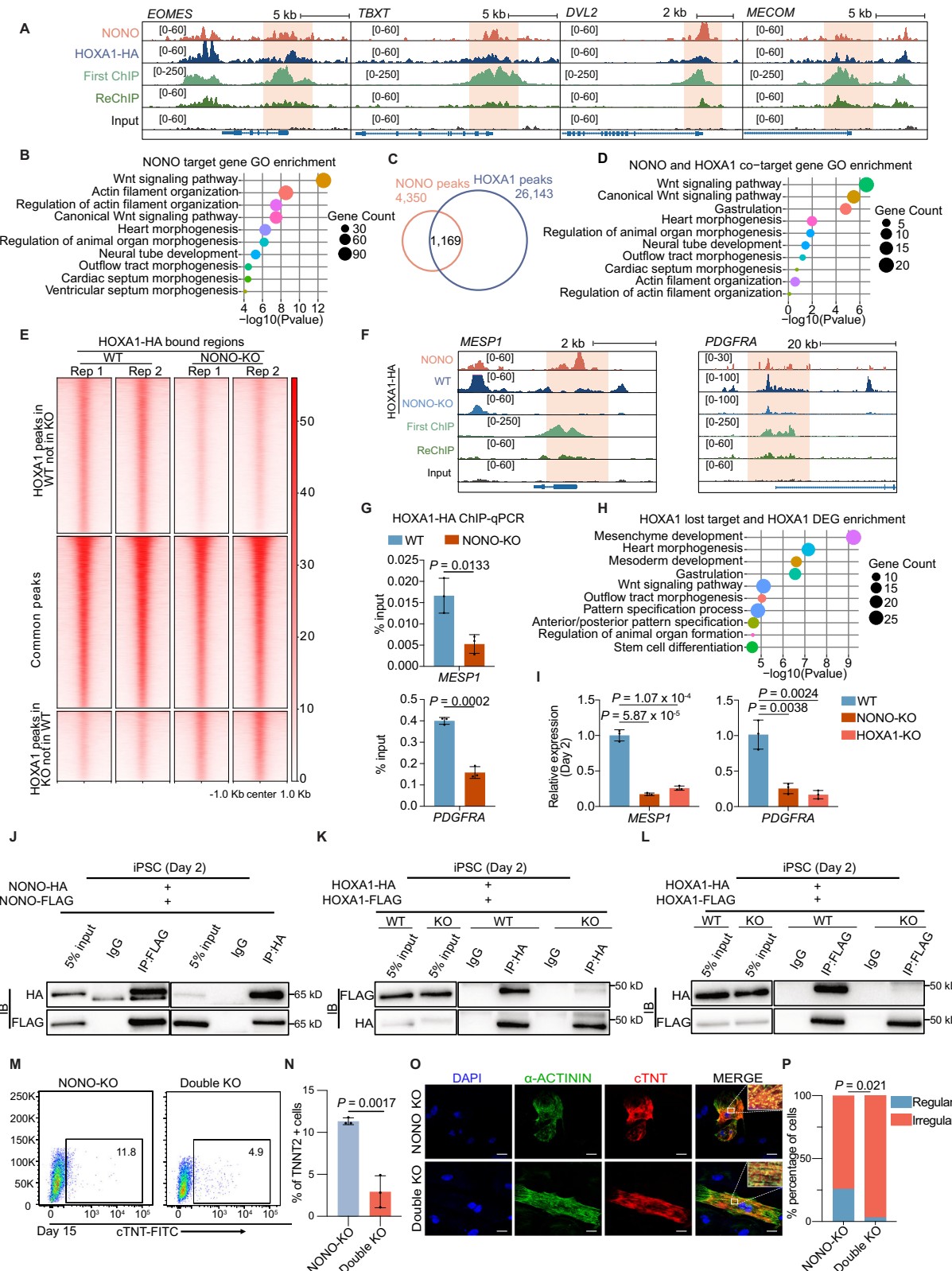

Finally, to confirm the functional significance of this cooperation, we generated *NONO-HOXA1* double-knockout (DKO) cells. Phenotypic analysis demonstrated that DKO cells exhibited significantly more severe defects in cardiac differentiation compared to either single KO line (Fig. 5M–P, Supplementary Fig. 1I, J). Together, these findings support a model where NONO promotes HOXA1 homodimerization to enable stable DNA binding and synergistic activation of the precardiac mesoderm program.

## The NONO-HOXA1 interaction is critical for Wnt/β-catenin activation

As demonstrated above, HOXA1-regulated genes dependent on NONO are significantly enriched in the Wnt signaling pathway. Among these genes, three were upregulated, and six were downregulated in both HOXA1-KO and NONO-KO cells (Fig. 6A, B). The expression patterns of key canonical Wnt signaling pathway genes, such as *WNT5A*, *WIF1*, and *GRB10*, were validated in cells on Day 2 of differentiation via qRT–PCR

**Fig. 5 | NONO cooperates with HOXA1 to activate the precardiac mesoderm gene. A** Genome browser visualization of ChIP-seq tracks at key cardiac gene loci. **B** GO biological process enrichment analysis of selected terms for genes bound by NONO in Day 2 differentiated Cells. **C** Venn diagram of NONO-enriched and HOXA1-HA enriched overlap peaks in Day 2 differentiated cells. **D** GO biological process enrichment analysis of selected terms for genes of NONO and HOXA1-HA co-binding. **E** Heatmaps of HOXA1 ChIP-seq signals in WT and NONO-KO cells. HOXA1 binding is significantly reduced in the absence of NONO, with shared and lost peaks shown. **F** Representative ChIP-seq tracks of HOXA1 binding at precardiac mesoderm loci (*MESP1* and *PDGFRA*) in Day 2 WT and NONO-KO cells. **G** ChIP-qPCR analysis showing reduced HOXA1 binding at target loci in Day 2 differentiated NONO-KO cells compared to WT. $n = 3$. **H** GO enrichment analysis of genes with lost HOXA1 binding in NONO-KO cells and differential expression (DEGs) in HOXA1-KO cells. **I** qRT-PCR analysis of HOXA1-regulated genes in Day 2 WT, NONO-KO, and HOXA1-KO cells. $n = 3$. **J** Co-immunoprecipitation (Co-IP) assessment of NONO

homodimerization in Day 2 cells expressing Dox-induced NONO-FLAG and NONO-HA. $n = 3$. **K, L** Co-IP analysis of HOXA1 homodimerization in Day 2 WT and NONO-KO cells co-expressing HOXA1-HA and HOXA1-FLAG, using anti-HA (**K**) and anti-FLAG (**L**) antibodies for immunoprecipitation, followed by immunoblotting with anti-FLAG and anti-HA antibodies. Note the impaired HOXA1 self-association in NONO-KO cells. $n = 3$. **M, N** Flow cytometry analysis (**M**) and quantification (**N**) of cTNT-positive cardiomyocytes in NONO-KO and NONO/HOXA1 Double-KO (DKO) at Day 15. $n = 3$. **O, P** Immunostaining (**O**) and quantification (**P**) of sarcomere organization (α-ACTININ, green; cTNT, red) in Day 15 NONO-KO and DKO cardiomyocytes. Scale bar, 15 μm. ($n = 4$; NONO-KO, 27 cells; DKO, 30 cells). Data are presented as mean values ± SD. *P* values were calculated using a two-tailed unpaired Student's *t* test, except for (**P**) (Fisher's exact test). $P < 0.05$ was considered significant. *n* represents independent biological replicates. Source data are provided as a Source Data file.

and ChIP–qPCR (Supplementary Fig. 10A, B). The findings suggested that NONO is critical for the HOXA1-mediated activation of key Wnt signaling components.

The transcription factor β-catenin is an important downstream effector of the canonical Wnt signaling pathway. Upon Wnt receptor activation, β-catenin stabilizes, enters the nucleus, and activates Wnt target genes together with TCF4[39]. To investigate the role of HOXA1 and NONO in regulating β-catenin, we assessed β-catenin expression in both the cytoplasmic and nuclear fractions of WT, HOXA1-KO, and NONO-KO cells on Day 2 of differentiation. Western blot analysis revealed that, compared with those in WT cells, cytoplasmic β-catenin levels in HOXA1-KO cells remained unchanged, but nuclear β-catenin levels were significantly lower, suggesting that HOXA1 plays a critical role in facilitating the nuclear accumulation of β-catenin (Fig. 6C and Supplementary Fig. 10C). Similarly, in NONO-KO cells, the cytoplasmic β-catenin level was unaffected, but the nuclear β-catenin level was significantly reduced. Partial recovery of nuclear β-catenin was observed in NONO-RE cells (Fig. 6D and Supplementary Fig. 10D). To determine whether this nuclear depletion stems from protein destabilization, we generated HOXA1-KO (Supplementary Fig. 1K, L) and NONO-KO HEK293T cell lines, and conducted CHX chase assays on nuclear fractions. The results demonstrated a marked acceleration in the turnover of nuclear β-catenin in both knockout lines compared to WT controls, despite stable cytoplasmic levels (Fig. 6E, F).

To evaluate the functional roles of HOXA1 and NONO in Wnt/β-catenin signaling in HEK293T cells, we used the TOP/FOP-Flash reporter assay stimulated with the Wnt activator CHIR99021 (CHIR). HOXA1 knockdown significantly reduced CHIR-induced TOP/FOP-Flash reporter activity, particularly at higher siRNA concentrations (Fig. 6G and Supplementary Fig. 10E). Conversely, HOXA1 overexpression enhanced reporter activity in a dose-dependent manner, confirming that HOXA1 positively regulates Wnt/β-catenin signaling (Fig. 6H). Similarly, NONO knockdown via siRNA reduced CHIR-induced reporter activity, indicating that NONO is essential for optimal Wnt signaling activation (Fig. 6I and Supplementary Fig. 10F). NONO overexpression also increased TOP/FOP-Flash reporter activity in a dose-dependent manner, further validating its role as a positive regulator of Wnt signaling (Fig. 6J).

Given that *NONO* depletion impairs the regulation of genes involved in the Wnt signaling pathway, a TOP/FOP luciferase assay was used to investigate the role of NONO in HOXA1-mediated Wnt activation. NONO overexpression enhanced HOXA1-driven Wnt signaling, suggesting that NONO facilitates the HOXA1-mediated activation of the pathway (Supplementary Fig. 10G). Further analysis revealed that the HOXA1-mediated activation of Wnt signaling is largely dependent on its HD, as mutating this domain significantly reduced luciferase activity (Fig. 6K). Additionally, the disruption of the HOXA1-NONO interaction resulted in decreased Wnt activity (Fig. 6L), emphasizing the importance of this interaction for maintaining Wnt signaling. To

evaluate whether stronger Wnt activation could rescue the differentiation defect in NONO-KO hiPSCs, we performed a CHIR99021 dosage–response assay. Increasing the Wnt stimulus to 10–12 μM at Day 0–2 restored CM differentiation, as reflected by higher TNNT2+ cell proportions (Fig. 6M, N) and increased expression of *TNNT2*, *MYH6*, and *MYH7* (Fig. 6O). In contrast, excessive CHIR exposure (14 μM) diminished differentiation efficiency, consistent with previously reported cytotoxic or lineage-disruptive effects of supraphysiological Wnt activation. These findings indicate that NONO deficiency elevates the threshold for Wnt-mediated mesoderm induction and that this defect can be mitigated by strengthening early Wnt activation.

Collectively, these findings indicate that NONO is indispensable for the HOXA1-mediated activation of the Wnt/β-catenin signaling pathway, emphasizing its cooperative role in regulating the Wnt pathway during CM differentiation (Fig. 6P).

## Discussion

The heart is the first organ to develop and function, and its proper formation is critical for preventing congenital anomalies and heart disease in mammals[40]. Genetic variants in *NONO* are commonly associated with various types of CHDs, underscoring the importance of our findings linking the important role of NONO to CM differentiation. In this study, we not only showed that NONO can form a molecular complex with HOXA1, which is consistent with our general understanding that NONO functions as a molecular chaperone to impact important biological processes, but also revealed that this interaction is critical for the canonical Wnt signaling pathway via a defined transcriptional pathway during CM differentiation. These findings establish a critical role for the NONO-HOXA1-Wnt signaling axis in early cardiac mesoderm development and CM differentiation and function.

The Wnt/β-catenin signaling pathway is recognized as a critical regulator involved in various stages of cardiac differentiation, including mesodermal cell specification and commitment, cardiac progenitor formation and differentiation, and CM maturation[41–44]. Despite the valuable insights gained from a series of studies using animal models, considerable challenges remain in understanding the regulation of Wnt/β-catenin signaling during human cardiac development[45,46]. Our study provides an interpretation of how the Wnt pathway is regulated by the NONO-HOXA1-mediated transcriptional pathway during early CM differentiation. Interestingly, HOXA1 has been delineated as a critical component of gene modules associated with the OFT and pSHF clusters during cardiac differentiation from hiPSCs, as evidenced by previous single-cell transcriptomic analyses[47,48]. Mice deficient in Hoxa1 exhibit severe cardiac defects, such as an interrupted aortic arch and tetralogy of Fallot[49]. In humans, mutations in *HOXA1* result in Bosley-Salih-Alorainy Syndrome (BSAS), the features of which include ocular motility disorders, sensorineural hearing loss, and cardiovascular malformations[31,50]. Recent research

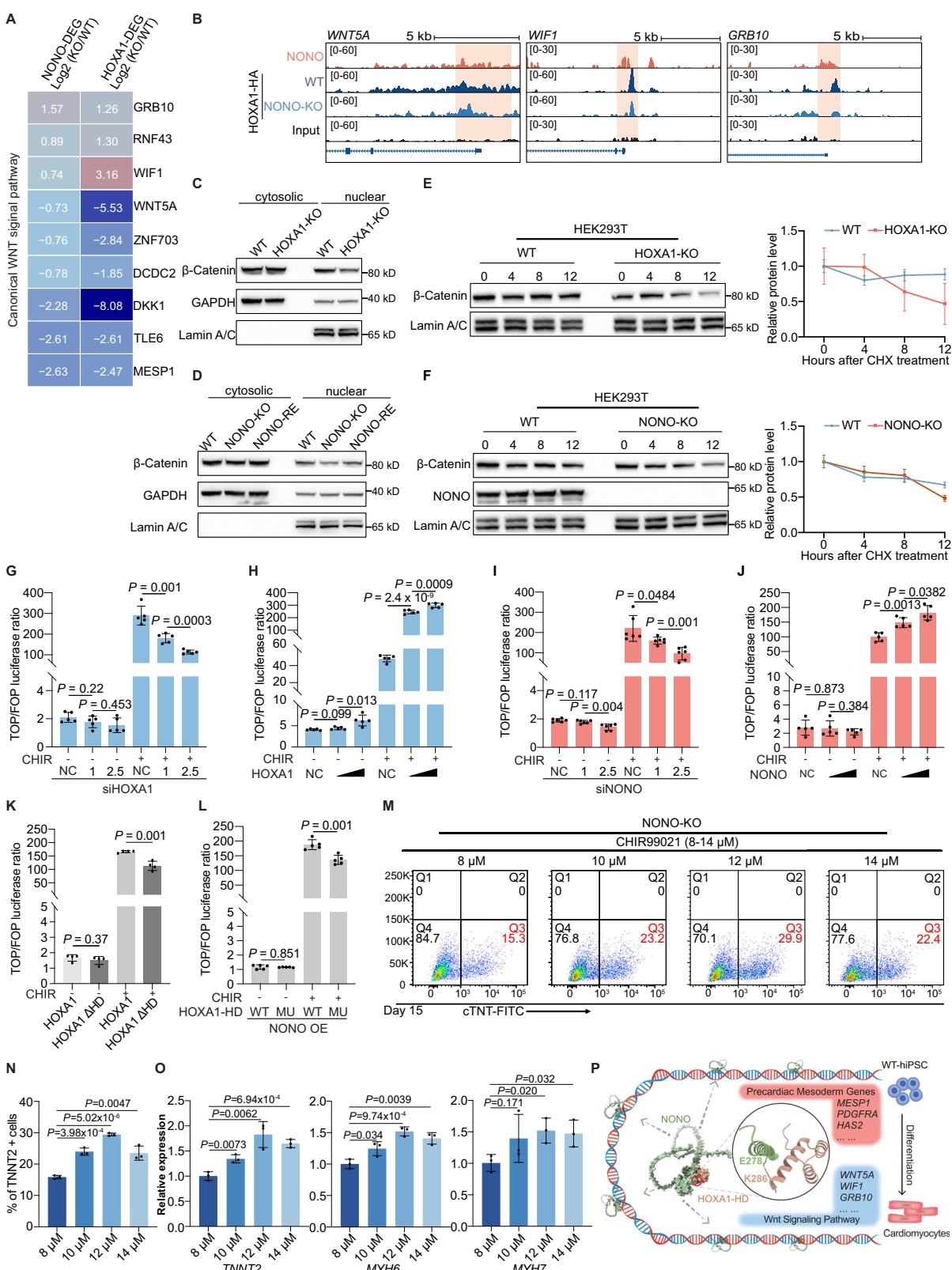

has also linked genetic variants in the histidine motif of *HOXA1* to the development of bicuspid aortic valves[32]. Our findings may also provide a key interpretation of how HOXA1, along with its cofactor NONO, regulates cardiac development, considering that Wnt canonical signaling plays major roles in the second heart field (SHF)[51].

Previous studies have highlighted the involvement of alternative splicing in sarcomerogenesis and its impact on heart development and function. These findings underscore the critical role of alternative splicing and its associated RNA-binding proteins (RBPs) in cardiac development, physiology, and the pathogenesis of heart diseases[52,53]. Analyses of DEGs, ASEs, and NONO-targeted genes revealed that 542 transcriptionally regulated genes were enriched in heart development pathways, whereas 48 genes exclusive to alternative splicing were not significantly enriched. The presence of 14 dually regulated genes

**Fig. 6 | HOXA1-NONO is required for regulating the WNT pathway. A** Heatmap showing log2-fold changes of differentially expressed genes (DEGs) among NONO-dependent and HOXA1-regulated WNT pathway genes in NONO-KO (left) and HOXA1-KO (right) cells. **B** Representative ChIP-seq tracks of WNT pathway genes showing HOXA1 binding in Day 2 WT cells, with reduced binding in NONO-KO cells. **C, D** Western blot analysis of cytoplasmic and nuclear β-CATENIN levels in Day 2 differentiated cells. WT vs. HOXA1-KO (**C**). WT, NONO-KO, and NONO-RE (**D**). n = 3. **E, F** Cycloheximide (CHX) chase assays and quantification of nuclear β-catenin stability in NONO-KO (**E**) and HOXA1-KO (**F**) cells. Protein levels were normalized to LAMIN A/C and expressed relative to the 0 h time point. n = 3. **G–J** TOP/FOP reporter assays in HEK293T cells treated with or without CHIR99021 (CHIR). Dose-dependent knockdown of HOXA1 (**G**) (siHOXA1; n = 5). Dose-dependent over-expression of HOXA1 (**H**) (n = 5). Dose-dependent knockdown of NONO (**I**) (siNONO; n = 6). Dose-dependent overexpression of NONO (**J**) (n = 5). **K** TOP/FOP reporter assay in cells overexpressing full-length HOXA1 or homeodomain-deleted mutant (HOXA1 ΔHD), treated with or without CHIR. n = 4. **L** TOP/FOP reporter assay in cells expressing WT or K286A mutants (MU) HOXA1, followed by NONO overexpression. Cells were treated with or without CHIR. n = 5. **M, N** Flow cytometry analysis (**M**) and quantification (**N**) of cTNT⁺ cardiomyocytes in Day 15 NONO-KO cultures treated with CHIR (8–14 μM) during Days 0–2. n = 3. **O** qRT-PCR analysis of TNNT2, MYH6, and MYH7 expression in Day 15 NONO-KO cardiomyocytes treated with CHIR (8–14 μM) during Days 0–2. n = 3. **P** Schematic model of the NONO-HOXA1-Wnt signaling axis in cardiomyocyte differentiation. Data are presented as mean values ± SD. P values were calculated using a two-tailed unpaired Student's t test. P < 0.05 was considered significant. n represents independent biological replicates. Source data are provided as a Source Data file.

further emphasizes the complexity of the NONO regulatory network. These results suggest that NONO transcriptional regulation is critical for early differentiation, whereas the NONO-mediated regulation of pre-mRNA splicing may have a greater impact on distinct cellular processes. Nevertheless, our findings highlight the dual role of NONO in transcriptional regulation and alternative splicing, with transcriptional activity playing a more prominent role in cardiac differentiation.

Beyond activating cardiac developmental pathways, our analysis revealed that NONO actively suppresses lineage-restricting factors. Established repressors such as EZH2 [54], REST [55], and HES1 [56] were aberrantly upregulated in NONO-KO cells (Supplementary Fig. 7D). This suggests a dual mechanism wherein NONO facilitates lineage commitment by synergistically activating cardiac identity genes while simultaneously dismantling transcriptional barriers imposed by progenitor-maintenance programs. Regarding the temporal dynamics of this regulation, we noted that MESP1 induction peaks at Day 3 in our hiPSC differentiation model (Supplementary Fig. 2B). While this timing is slightly delayed compared to mesodermal specification in vivo, it is consistent with established kinetics in pluripotent stem cell-based differentiation systems[37,57,58]. We acknowledge that in vitro models cannot fully recapitulate the precise chronometry of embryonic development, nor do they possess the complex spatial architecture of native organogenesis. However, the hiPSC system offers an ideal, controlled platform for dissecting molecular mechanisms. Therefore, validating our findings in vivo is a critical next step. Future studies using NONO-deficient mice will determine whether the molecular defects observed here translate into anatomical malformations during heart development.

We demonstrated that NONO is important for proper CM differentiation, which was supported by two major observations, including transcriptomic analyses: NONO-KO hiPSCs exhibited significantly lower differentiation efficiency and disrupted CM function, and NONO-RE hiPSCs exhibited partially rescued phenotypes (Fig. 2). These observations align with previous studies that employed other models. For example, in an H9c2 rat CM differentiation model and a mouse aortic vascular smooth muscle cell model, a reduction in the level of NONO reduced cellular proliferation, adhesion, and sarcomere organization[59–61]. Interestingly, Nono KO in mESCs enhanced pluripotency and self-renewal properties[62]. However, our NONO-KO hiPSCs had largely normal self-renewal or pluripotency abilities. This discrepancy may be attributed to species-specific differences and intrinsic differences between mESCs and hiPSCs, such as distinct transcriptional programs and epigenetic regulation. Future analyses comparing the mESC and hiPSC transcriptomes during CM differentiation may resolve this apparent discrepancy. In addition, our current study focused on early CM differentiation; we will next focus on how the NONO-HOXA1 complex regulates CM function, which we believe is highly relevant to the regulation of CM homeostasis and the maintenance of normal cardiac function.

In summary, our study provides evidence that NONO directly interacts with HOXA1 and that NONO-HOXA1 is a critical transcriptional regulatory complex during CM differentiation. Our findings demonstrated that NONO-HOXA1 transcriptionally regulates Wnt signaling by modulating the expression of critical genes involved in Wnt signaling. These findings highlight the importance of the NONO-HOXA1-Wnt signaling axis in early cardiac mesoderm development and provide insights into the molecular mechanisms underlying CHDs.

## Method

### hiPSC culture and CM differentiation

Research involving human induced pluripotent stem cells (hiPSCs) was performed in strict accordance with the 2021 ISSCR Guidelines for Stem Cell Research and Clinical Translation. Human Induced pluripotent stem cells (WT-hiPSCs) derived from normal individuals were provided by Professor Ning Sun from the Department of Pathophysiology, Basic Medical College, Fudan University. This established cell line was generated from a skin punch biopsy of a healthy donor under ethical approval from the Human Research Ethical Review Board of Zhongshan Hospital of Fudan University, and informed consent was obtained from the donor as previously described[63]. hiPSCs were cultured on Matrigel-coated plates using mTeSR™ plus medium (Stemcell Technologies, 100-0276). Upon reaching ~80% confluence, the cells were split at a ratio of 1:5 using Accutase (Sigma, A6964). We utilized a previously established protocol involving WNT activation and inhibition to differentiate hiPSCs into CMs[21]. Briefly, hiPSCs were seeded at 80% confluence, treated with B27 minus insulin (Thermo, A1895601)/RPMI1640 (Thermo, 11875119) plus 7 μM CHIR99021 (Selleck, S1263). Following two Days of CHIR99021 treatment, the medium was switched to B27 minus insulin/RPMI1640. After 24 hours, the medium was replaced with B27 minus insulin/RPMI1640 containing 5 μM IWR1 (Selleck, S7086) for 2 Days. Subsequently, IWR1 was removed, and differentiation continued in CDM3 medium containing RPMI640, 50 mg/ml BSA, and 213 mg/ml ascorbic acid. Beating CMs was observed by Day 8.

### HEK293T cell culture and transient transfection

HEK293T cell line was obtained from ATCC (ATCC: ACS-4500) and cultured in Dulbecco's modified Eagle's medium (DMEM) (Gibco, 11995-065) supplemented with 10% fetal bovine serum (FBS) (Gibco, 10099-141 C) and 1% Penicillin-Streptomycin (Gibco, 15070-063). Transient transfection was performed using jetPRIME® (Polyplus, 10100046), following the manufacturer's protocol.

siRNAs were obtained from QianMo Biotechnology Co., Ltd. (Shanghai, China): NONO siRNA target sequence, AGGTCATGCTAAT-GAGACA and HOXA1 siRNA target sequence, CTACGCGTTAAATCAG-GAA. Transfection of these siRNAs was carried out using jetPRIME® (Polyplus, 10100046) according to the manufacturer's protocol.

### Generation of CRISPR-mediated gene knockout hiPSC and HEK293T cell lines

Single-guide RNAs (sgRNAs; sequences provided in Supplementary Data 1) were designed using DeepHF (https://www.deephf.com/[64]) and

subsequently cloned into the EpiCRISPR vector, kindly provided by Dr. Yongming Wang. All sgRNA oligonucleotides were synthesized by Tsingke Biotechnology Co., Ltd. (Beijing, China). The EpiCRISPR-gRNA construct was transfected into hiPSCs or HEK293T cell lines using the Lonza 4D Nucleofector system (Lonza, AAF-1002X). Gene knockout (KO) clones were selected with puromycin (Thermo, A1113803) for one week, followed by screening through genomic sequencing, qRT-PCR and western blot analysis.

### Generation of doxycycline-inducible NONO and HOXA1 expression in hiPSC lines

NONO-FLAG (pcDNA3.1) and HOXA1-HA (pcDNA3.1) plasmids were already developed in our laboratory. A PiggyBac transposon system kindly provided by Professor Ning Sun was used to establish doxycycline-inducible NONO expression in NONO-KO hiPSC lines and doxycycline-inducible HOXA1 overexpression hiPSC lines. The cDNA encoding the FLAG-NONO/HOXA1-HA tag fusion protein was sub-cloned downstream of the TRE3G promoter using the ClonExpress Ultra One Step Cloning Kit V2 (Vazyme, C116-01), resulting in the construction of the PB-TRE-NONO-FLAG, PB-TRE-NONO-HA, PB-TRE-HOXA1-FLAG, and PB-TRE-HOXA1-HA expression vectors. These Pig-gyBac expression plasmids were co-transfected together with the rtTA and PiggyBac transposase vectors (2:2:1) into WT and NONO-KO hiPSCs using the Lonza 4D Nucleofector system. The electroporated cells were subjected to 1 week of combined puromycin (Selleck, S7471) and G418 (Gibco, 10131) selection. Positive clones expressing NONO and HOXA1-HA in response to doxycycline induction were jointly screened using western blot analysis. This strategy enabled doxycycline-inducible reactivation of NONO in NONO-KO hiPSCs, overexpression of NONO-FLAG and NONO-HA in WT-hiPSCs, and inducible expression of HOXA1-FLAG and HOXA1-HA in both WT and NONO-KO backgrounds.

### Immunofluorescence staining

Individual hiPSC clones and single beating CMs, dissociated using TrypLE Select (Thermo, 12563029), along with HEK293T cells dis-sociated with 0.25% Trypsin-EDTA (Thermo, 25200056), were plated on 35 mm imaging dishes (ibidi, 81156) pre-coated with Matrigel. The cells were fixed with 4% paraformaldehyde (Solarbio, P1110) for 15 minutes (min), permeabilized with 0.02% Triton-X-100 for another 15 min, and blocked with 5% BSA for 1 hour prior to staining. For pluripotency assessment, hiPSC colonies were stained with antibodies against OCT4 (Abcam, ab181557) at a 1:500 dilution and SSEA4 (Abcam, ab16287) at a 1:200 dilution. CM differentiation was assessed using antibodies against TNNT2 (Thermo, MA5-12960) at a 1:200 dilution and ACTN2 (Proteintech, 14221-1-AP) at a 1:100 dilution. For coloci-zation studies of NONO and HOXA1, staining involved antibodies against NONO (Santa Cruz, sc-376865) at a 1:200 dilution and HA tag (Abcam, ab9110) at a 1:500 dilution. Following primary antibody application, all cells were incubated with Alexa Fluor-conjugated sec-ondary antibodies (Thermo, A32731 and A32723) at a 1:1000 dilution at room temperature for 1 hour. Cells were then counterstained with DAPI for 15 min at room temperature. The labeled cells were visualized using a Leica DMi8 fluorescence microscope.

### Alkaline phosphatase staining (AP)

hiPSCs were seeded on a six-well plate coated with Matrigel and cul-tured for 2–3 Days. AP staining was performed using the Quantitative Alkaline Phosphatase ES Characterization Kit (Millipore, SCR004), following the manufacturer's protocol.

### Calcium transient imaging

Day 15 differentiated CMs were loaded with 5 μM Cal-520® AM (AAT Bioquest, 21130) and 0.02% Pluronic F-127 (AAT Bioquest, 20053), and then incubated at 37 °C for 15 min in Tyrode's solution (Solarbio, T1420).

After that, CMs were washed three times using Tyrode's solution. The calcium transients of CMs were recorded using a Leica DMi8 fluores-cence microscope (10 ms per scan, 30 s). The LAS X Science Microscope Software (Leica) and Excel were used for the transition of the calcium image into data. We developed an R script (https://doi.org/10.5281/zenodo.18059608) to detect inflection points for the data of calcium transition, followed by calculation and statistical analysis based on the calcium transient parameter schematic shown in Fig. 1H.

### Flow cytometry assay

Single beating CMs, dissociated using TrypLE Select, Dissociated cells were washed with PBS and Perm/Wash Buffer (BD Biosciences, 554715), then fixed and permeabilized using Fixation/Permeabilization Solution for 60 min. Cells were incubated for 1 hour with either anti-cardiac troponin T (cTnT) primary antibodies or isotype-matched controls. For additional lineage analyses, dissociated cells were separately stained with antibodies against PDGFRA (R&D Systems, AF1062), NKX2-5 (CST, 8792), or GATA4 (Santa Cruz, sc-25310) using the same fixation and permeabilization conditions. After washing, cells were stained with secondary antibodies (Proteintech, SA00003-1, SA00008-9 and SA00008-2) for 30 min, then washed, resuspended, and analyzed using a FACS Calibur flow cytometer (BD Biosciences). Data were processed with FlowJo software v10. All data were analyzed using FlowJo software (BD). The gating strategies used for data analysis are shown in Supplementary Fig. 11.

### Transmission electron microscopy

CMs were immediately fixed with 2.5% glutaraldehyde solution for 15 min at room temperature, CMs underwent dehydration in escalating ethanol concentrations: 60%, 70%, 80%, 90%, and 95%, each for 24 hours, followed by two treatments in 100% ethanol, each lasting 12–24 hours. The samples were then embedded in a mold with T7200 resin and cured under light for 12 hours to polymerize. Sectioning involved adhering the embedded blocks to slides using T4000 glue, smoothing with 1200-grit sandpaper, and affixing to a top slide with T7210 precision adhesive using a parallel sectioning device. This configuration was exposed to blue light for 10 min before disassembly and then vacuum-mounted on a hard tissue microtome to cut 200 μm thick sections. Finally, these sec-tions were polished down to 30 μm thickness using a hard tissue grinder and analyzed using a TEM (HT7800; HITACHI).

### Quantitative reverse transcription real-time polymerase chain reaction (qRT-PCR)

Total RNA was extracted using Trizol (Thermo, 15596018) according to the manufacturer's protocol. Complementary DNA (cDNA) was pre-pared using the PrimeScript™ RT reagent Kit (Takara, RR036A) with 1 μg RNA. qRT-PCR was performed using the TB Green® Premix Ex Taq™ (Takara, RR420A) on a QuantStudio 3 system (Applied Biosys-tems). Relative mRNA expression was normalized to the GAPDH expression level. Primer sequences are provided in Supplementary Data 1, and all primers were synthesized by Tsingke Biotechnology Co., Ltd. (Beijing, China).

### Bulk RNA-seq and alternative splicing analysis

RNA-seq library preparation was performed according to the manu-facturer's instructions (Vazyme, TR503-01). The libraries underwent high-throughput sequencing conducted by Haplox company (Jiangxi, China) using the Illumina NovaSeq 6000 platform, ensuring three biological replicates.

For RNA-seq data analysis, sequencing reads in FASTQ format were first processed using Trim Galore (v0.6.4) with the parameters '--paired --illumina' to remove low-quality reads and adapter sequen-ces. The trimmed reads were subsequently aligned to the human reference genome (UCSC hg38) using TopHat (v2.1.1) under default settings. Gene expression quantification was performed using

Cufflinks (v2.2.1) with default parameters, which assigned the mapped reads to human gene annotations and generated expression abundances in FPKM (Fragments Per Kilobase of transcript per Million mapped reads). The FPKM values were then normalized to TPM (Transcripts Per Kilobase Million) to enable cross-sample comparison of gene expression levels. Differential expression analysis was carried out using Cuffdiff within the Cufflinks suite, which computed fold-changes and $P$ values for each gene. The analysis included three biological replicates for Days 2, 5, 9, and 15, with differentially expressed genes defined as those exhibiting a fold-change greater than 1.5 and a P value less than 0.05. To identify alternative splicing events, aligned reads were analyzed using rMATS, with significant splicing changes defined by $P$ value < 0.05 and junction read counts ≥5. Functional enrichment analysis, including GO term and KEGG pathway analysis, was conducted on the differentially expressed genes using the R package clusterProfiler (v3.14.0) from Bioconductor.

## 9-square plot analysis

To compare the changes in differentially expressed genes (DEGs) between KO vs WT and KO vs Rescue, we first identified DEGs in both comparisons with a cutoff of 1.5-fold change. To visually represent these DEG changes in a scatter plot, we categorized the genes into a 9-square matrix based on their regulation patterns. Each point in the matrix represents a gene, and genes were assigned to 9 groups (A–I) based on their up- or downregulation according to the 1.5-fold change threshold. Log2(fold-change) values greater than 4 or less than −4 were scaled to 4 or −4, respectively. Group C contains genes that are upregulated in both KO vs WT and KO vs Rescue, while group G contains genes that are downregulated in both comparisons. This matrix allows for a direct comparison of DEG changes between KO vs WT and KO vs Rescue, as shown in the scatter plot, where the different groups are visually distinguished by their respective fold-change patterns.

## Western blot (WB)

Whole cell lysates were prepared using RIPA buffer (Thermo, 89901) containing protease inhibitor cocktail. Lysates were boiled with 1× Sample buffer (GenScript, M01015) for 5 min and separated by 4–20% SurePAGE™ Bis-Tris Gels (GenScript, M00655), followed by transfer to PVDF membranes (Millipore, IPVH00010). The Membranes were blocked for 1 hour at room temperature in 1× Tris Buffered Saline containing Tween-20 (TBST) (Beyotime, ST673) with 5% non-fat dry milk (BD biosciences, 232100), then incubated overnight at 4 °C with primary antibodies against NONO at a dilution 1:1000, HA at a dilution 1:5000, GAPDH (Proteintech, 10494-1-AP) at a dilution 1:10000, GFP (Abcam, ab290) at a dilution 1:2000, GST (Proteintech, 10000-0-AP), β-catenin (Abcam, ab32572) at a dilution 1:3000, and Lamin A/C (CST, 4777) at a dilution 1:2000. The next Day, membranes were washed three times of 10 min each with TBST, and incubated 2 hours at room temperature with HRP-conjugated secondary antibodies (CST, 7074/7076) at 1:3000 dilution. The bands were incubated with SuperSignal West Pico PLUS (Thermo, 34580) for 2 min and detected using Che-miDoc Imaging System (Biorad XRS + ). The intensity of each band was quantified using Image Lab software.

## CHX chase assays

HEK293T cells were seeded in six-well plates and transfected with 1 µg of plasmids using JetPrime reagent. To assess protein stability, cells were treated with 200 µg/mL CHX (Sigma-Aldrich, 01810) 24 hours post-transfection to inhibit protein synthesis. ImageJ software was used to estimate protein abundance.

## Subcellular fractionation

Subcellular fractionation was prepared using NU-PER Nuclear and Cytoplasmic Extraction Reagents (Thermo, 78835) according to the manufacturer's protocol. The entire process was performed at 4 °C.

The cells were harvested with TryPLE Select and then centrifuged for 5 min. Cells were washed with PBS 2 times and incubated with CER I buffer containing protease inhibitor cocktail for 10 min, then incubated with CER II for 1 min. Centrifuge the tube to get the cytoplasmic extraction, then suspend the insoluble fraction with NER containing protease inhibitor cocktail. Vortex for 15 seconds every 10 min, for a total of 40 min, then sonicate in low mode (15 s on, 15 s off, 15 cycles) using Biorupter® Plus sonication device (Diagenode, B01020001). Centrifuge the tube to get the nuclear extraction. The Nuclear and Cytoplasmic proteins were then detected by WB.

## Co-immunoprecipitation

For 293 T cells, cells overexpressing HOXA1 and NONO were lysed in a 10 cm dish using 1 ml IP lysis buffer containing protease inhibitor cocktail and incubated on ice for 20 min. Similarly, Day 2 differentiated cells with DOX-induced HOXA1-HA overexpression ($7–8 \times 10^6$ cells) were treated with the same lysis protocol. The lysates were then centrifuged at $12,000 \times g$ for 20 min. Supernatants were precleared with 15 µl each of Dynabeads Protein G (Thermofisher, 10004D) and Protein A (Thermofisher, 10002D), rotated at 4 °C for 1 hour to reduce non-specific binding, and subsequently cleared beads were removed with a DynaMag Magnet. 50 µL of each supernatant was saved as input. The remaining supernatant was incubated either with NONO and rabbit control IgG or HA and rabbit control IgG (4 µg of each antibody), rotated at 4 °C overnight (16 hours). Following antibody binding, 15 µl each of Dynabeads Protein G and Protein A were added and incubated for an additional 2 hours at 4 °C. The beads were washed three times with IP lysis buffer and then boiled at 95 °C for 10 min in 50 µl of 2× SDS sample buffer. The supernatant from this final boiling step was collected for further analysis.

## LC−MS/MS analysis and NONO protein complex identification

Wild-type and NONO-KO hiPSCs were differentiated for two Days, followed by immunoprecipitation with a NONO antibody. Protein complexes were eluted and analyzed by mass spectrometry. NONO-KO Day 2 differentiated cells served as a negative control. ($n = 1$). Proteins bound to the IP beads were eluted using SDT lysis buffer (4% SDS, 100 mM DTT, 100 mM Tris-HCl, pH 8.0). Samples were boiled for 3 min and further disrupted by ultrasonication. Insoluble material was removed by centrifugation at $16,000 \times g$ for 15 min, and the supernatant containing solubilized proteins was collected. Protein digestion was performed using the filter-aided sample preparation (FASP) method as previously described[65]. Briefly, samples were processed in urea buffer containing iodoacetamide to block reduced cysteines. Proteins were then digested with 2 µg trypsin (Promega) overnight at 37 °C. Peptides were collected by centrifugation at $16,000 \times g$ for 15 min and desalted using C18 StageTips prior to LC−MS/MS analysis.

LC−MS experiments were performed on a Q Exactive HF-X mass spectrometer that was coupled to Easy nLC1200 (Thermo Scientific). The peptide was first loaded onto a trap column (100 µm × 20 mm, 5 µm, C18, Dr. Maisch GmbH, Ammerbuch, Germany) in buffer A (0.1% Formic acid in water). Reverse-phase high-performance liquid chromatography (RP-HPLC) separation was performed using a self-packed column (75 µm × 150 mm; 3 µm ReproSil-Pur C18 beads, 120 Å, Dr. Maisch GmbH, Ammerbuch, Germany) at a flow rate of 300 nL/min. The RP − HPLC mobile phase A was 0.1% formic acid in water, and B was 0.1% formic acid in 95% acetonitrile. The gradient was set as follows: 2%–4% buffer B from 0 min to 2 min, 4% to 30% buffer B from 2 min to 47 min, 30% to 45% buffer B from 47 min to 52 min, 45% to 90% buffer B from 52 min to 54 min, 90% buffer B kept till to 60 min. MS data were acquired using a data-dependent top20 method dynamically choosing the most abundant precursor ions from the survey scan (350–1800 m/z) for HCD fragmentation. A lock mass of 445.120025 Da was used as an internal standard for mass calibration. The full MS scans were acquired at a resolution of 60,000 at m/z 200, and 15,000 at m/z 200 for the

MS/MS scan. The maximum injection time was set to 50 ms for MS and 25 ms for MS/MS. Normalized collision energy was 28, and the isolation window was set to 1.6 Th. Dynamic exclusion duration was 30 s.

The MS data were analyzed using MaxQuant software version 2.0.1.0[66]. MS data were searched based on the UniProt-Reference proteome-*Homo sapiens* (Human) [96060-81791-20230317.fasta (https://www.uniprot.org/proteomes/UP000005640, 81791 total entries). Settings were set as below: Enzyme-trypsin; Max Missed Vleavages-2; Precursor Tolerance (Main search)−4.5 ppm; Precursor Tolerance (First search)−20ppm; Variable modifications-Oxidation(M), Acetyl(Protein-N-term); Database pattern-Target-Reverse; PSM FDR-0.01; Protein FDR-0.01; Site FDR-0.01.

## GST Pull down

NONO GST-fusion constructs were expressed in BL21 Escherichia coli, which were grown to an appropriate optical density and induced with 0.5 mM IPTG (Selleck, S6826) at 22 °C for 4 hours. Cell pellets were resuspended in 10 mL of lysis buffer (10 mM Tris-HCl pH 7.5, 150 mM NaCl, 0.5 mM EDTA, 0.5% NP-40, 0.09% sodium azide) and sonicated for 30 min (10 seconds on, 10 seconds off) at 4 °C. The lysates were centrifuged at 12,000 × g for 20 min. The resulting supernatants were mixed with supernatants from HOXA1-HA overexpressing HEK293T cell lysates and incubated with rotation at 4 °C for 16 hours. To prepare for affinity purification, 25 μL of GST Trap Agarose (Proteintech, sta) was added to a microcentrifuge tube along with 500 μL of dilution buffer (10 mM Tris-HCl, pH 7.5, 150 mM NaCl, 0.5 mM EDTA, 0.018% sodium azide), centrifuged at 2500 × g for 5 min at 4 °C, and the supernatant was discarded. The equilibrated agarose was then introduced to the mixed proteins and incubated with rotation at 4 °C for 2 hours. The agarose was washed three times with 500 μL of wash buffer (10 mM Tris-HCl, pH 7.5, 150 mM NaCl, 0.5 mM EDTA, 0.05% NP-40, 0.018% sodium azide), each time centrifuging at 2500 × g for 5 min at 4 °C and discarding the supernatant. Finally, the agarose was resuspended in 50 μL of 2× SDS sample buffer and boiled at 95 °C for 10 min. The proteins were then detected by WB.

## Structure modeling with AlphaFold2

The interaction between NONO (Uniprot accession, Q15233) and HOXA1 homeodomain (HOXA1-HD, amino acid 229-288 of Uniprot accession P49639) was modeled utilizing AlphaFold-multimer (v2.3.2) with default parameters. The complex formed by NONO and HOXA1-HD was then visualized and analyzed using Chimera (v1.18).

## Chromatin immunoprecipitation (ChIP) and ChIP-seq

$7-8 \times 10^6$ Day 2 differentiated cells were crosslinked with 1% formaldehyde for 10 min at room temperature, then quenched with 0.125 M glycine for 5 min. Cells were washed three times with cold PBS and harvested in DNA LoBind Tubes (Eppendorf, 0030108035/ 0030108051). Sonication was performed using a Bioruptor (Diagenode, B01020001) on high setting (30 sec on, 30 sec off) for 30 min in high salt lysis buffer (20 mM Tris-HCl pH 8.0, 150 mM NaCl, 2 mM EDTA, 0.05% SDS, 1% Triton-X-100) containing a protease inhibitor cocktail. Chromatin samples were precleared with 15 μL each of Dynabeads Protein G and Protein A, and rotated at 4 °C for 1 hour. Precleared samples were then incubated with 4 μg of specific antibody at 4 °C for 16 hours. After antibody binding, 15 μL each of Dynabeads Protein G and Protein A were added and incubated for an additional 2 hours at 4 °C. The chromatin-antibody complexes underwent four washes with high salt buffer, two washes with low salt buffer (10 mM Tris-HCl pH 8.0, 250 mM LiCl, 1 mM EDTA, 0.5% NP-40, 0.5% sodium deoxycholate), and one wash with TE buffer (50 mM Tris-HCl pH 8.0, 10 mM EDTA, 50 mM NaCl). Complexes were eluted in elution buffer (50 mM Tris-HCl, pH 8.0, 10 mM EDTA, 1% SDS) on a Thermal Mixer (Thermo, 13687712) at 65 °C for 4 hours, followed by treatment with 10 μg RNase and 10 μg Proteinase K at 55 °C for 1 hour on the Thermal Mixer. DNA was purified using a QIAquick PCR Purification Kit (QIA-GEN, 28006). For Re-ChIP assays, the starting material was scaled to $2 \times 10^7$ cells. Chromatin was first immunoprecipitated with 12 μg of anti-HA antibody and competitively eluted under native conditions using HA peptide (1 mg/mL) (MCE, HY-P0239). Subsequently, 30% of the eluate was retained as the primary ChIP fraction, while the remaining 70% was diluted and subjected to incubation with 12 μg of anti-NONO antibody overnight, and then captured with Dynabeads. The purified DNA library preparation was constructed using VAHTS Universal Plus DNA Library Prep Kit for Illumina (Vazyme, ND617) in accordance with the manufacturer's instructions. Libraries were sequenced by the Haplox company (Jiangxi, China) using Illumina NovaSeq 6000.

## ChIP-seq data analysis

Sequencing reads underwent trimming with Trim_galore (v0.6.4_dev) and were subsequently aligned to human genomes using Bowtie2 (v2.2.5)[67]. PCR duplicate reads were eliminated using the rmdup command of samtools (v1.7)[68]. To visualize ChIP signals in the UCSC Genome Browser, genomic coverage bedGraph files were generated using the bamCoverage in deeptools (v3.0.2)[69]. Peak calling was performed using MACS2[70] with the following parameters: -q 0.1 -f BAMPE --extsize=500 --bw 300 for NONO, and -f BAMPE --extsize=300 --bw 300 for HOXA1. Heatmaps and aggregate plots were generated using deeptools computeMatrix for data organization, followed by plotHeatmap and plotProfile for visualization, respectively. The genomic distribution of ChIP-seq peaks was assessed using the R package ChIPseeker (v3.10)[71], with promoter regions defined from TSS-2kb to TSS + 2 kb.

## Single-cell RNA-sequencing and preprocessing

iPSCs were differentiated into CMs for scRNA-seq. Differentiation efficiency was evaluated by flow cytometry. Differentiated cells collected on Days 2, 5, and 15 were dissociated using Accutase and neutralized with FBS in DMEM. For each time point, three biological replicates were obtained from both WT and KO groups. After centrifugation at 1000 × g for 5 min, cell pellets were resuspended in DPBS containing 0.04% bovine serum albumin and immediately submitted for scRNA-seq. Single-cell libraries were generated using the Chromium GEM-X Single-Cell 3′ Kit v4 (10x Genomics) according to the manufacturer's instructions, and sequenced on an Illumina Novaseq X to a mean depth of at least 20 K reads per cell.

Sequencing reads were aligned to the human reference genome (GRCh38), and gene-by-cell count matrices were generated. Downstream analysis was performed using Seurat (v5.3.0)[72] and the Pegasus pipeline (v1.9.0)[73]. Quality control included removing cells with low gene counts (< 500) or elevated mitochondrial transcript proportions (>10%) to exclude low-quality or dying cells. Gene expression matrices were then normalized using SCTransform, and the top 3000 highly variable genes were selected. The Seurat object was subsequently converted to an AnnData object for further processing with Pegasus. Principal component analysis was performed, and the top 50 principal components were used to construct a shared nearest-neighbor graph. Batch effects across samples were mitigated using scVI[74] implemented in the Pegasus framework, followed by dimensionality reduction with UMAP. Cell type identities were assigned based on cluster-specific marker genes, supported by previously published annotations.

Cell differentiation potential was estimated using the CytoTRACE kernel implemented in the CellRank package, applying the CytoTRACEKernel().compute_cytotrace() function with default parameters[75]. Pseudotime trajectories for mesendoderm, cardiac progenitor, and CM populations were reconstructed using Monocle2[24]. Highly variable genes were used for dimensionality reduction with DDRTree, after which cells were ordered along the inferred trajectories to assign pseudotime values.

## TOP/FOP reporter assay

HEK293T cells were plated in 12-well plates and transfected with the indicated plasmids or siRNAs for 24 hours, followed by a second transfection with either TOP-FLASH or FOP-FLASH reporter plasmids together with pRL-TK. After another 24 hours, cells were dissociated and re-plated into 96-well plates, distributing the contents of each well from the 12-well plate into six wells of the 96-well plate. The cells were then treated with either DMSO or 7 μM CHIR99021, a WNT/β-catenin activator, for 24 hours. Luciferase activity was measured using the Dual-Luciferase Reporter Assay System (Promega, E1960) according to the manufacturer's protocol, and the results were detected using a Varioskan LUX Multimode Microplate Reader (Thermo, VL0000D0). The average luciferase activity from the six wells was calculated, normalized to Renilla luciferase activity, and expressed as the TOP/FOP ratio to determine relative Wnt activity. All transfection experiments were conducted with at least three biological replicates.

## Statistical analysis

Data are presented as mean ± standard deviation (SD) or mean ± standard error of the mean (SEM), as indicated in the figure legends. All quantitative experiments were performed with at least three independent biological replicates. An unpaired two-tailed Student's $t$ test and Fisher's exact test were used to calculate significant differences between the two groups. $P$ values < 0.05 were considered to indicate a statistically significant result. All analyses were performed using GraphPad Prism version 9.0.

## Reporting summary

Further information on research design is available in the Nature Portfolio Reporting Summary linked to this article.

## Data availability

The scRNA-seq data generated in this study have been deposited in the GEO database under accession code GSE315814. The RNA-seq data generated in this study have been deposited in the GEO database under accession code GSE295273. The ChIP-seq data generated in this study have been deposited in the GEO database under accession code GSE295270. The mass spectrometry proteomics data have been deposited to the ProteomeXchange Consortium via the PRIDE[76] partner repository with the dataset identifier PXD072401. Source data are provided with this paper.

## Code availability

Custom R scripts for automatic detection and quantification of calcium transient inflection points can be found in the following Zenodo repository: https://doi.org/10.5281/zenodo.18059608.

## Material availability

All unique biological materials generated in this study, including the NONO-KO and HOXA1-KO hiPSC lines, plasmid constructs, and sgRNA vectors, are available from the corresponding author upon reasonable request.

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

## Acknowledgements

This work was supported by the Grants from National Key Research and Development Program of China (2021YFC2701000, 2016YFC1000500 to G.H.), the National Natural Science Foundation of China (82270312 to G.H., 32070653 to F.W. and 82200264 to S.L.), Shanghai Municipal Science and Technology Major Project (2017SHZDZX01 to F.W.), AI for Science Foundation of Fudan University (FudanX24A1052 to F.W.), Science and Technology Projects of Xizang Autonomous Region, China (XZ202401ZY0042 to W.S.), Fujian Provincial Health and Health Technology Program (2024GGB26 to X.H.), China Postdoctoral Science Foundation (2025M781912 to Z.F.) and the CAMS Innovation Fund for Medical Sciences (2019-I2M-5-002 to G.H.). The data analysis was supported by the Medical Science Data Center of Fudan University. We thank Professor Ning Sun for providing the WT-hiPSCs and PiggyBac transposon system constructs, and Professor Yongming Wang for providing the epi-CRISPR system constructs. We also thank Dr. Ran Yang for valuable suggestions regarding the scRNA-seq analysis.

## Author contributions

G.H., F.W., and W.S. conceived and designed the study, Z.F. and Y.G. interpreted the data and wrote the manuscript. Z.F., H.G., S.S., and W.N. generated the KO and OE cell lines, conducted the hiPSC CM differentiation work experiments. Z.F., Y.G., X.H., S.L., S.S., and C.T. performed co-IP and pull-down experiments. Z.F., Y.G., and H.G. performed scRNA-seq experiments and bioinformatic analysis. Z.F., F.W., S.M., M.W., J.W., and Z.G. performed ChIP-seq experiments and bioinformatic analysis. S.Y.L., Y.Q.L., and Q.Z. provided valuable suggestions for data statistical analysis. X.M., Y.L., and W.N.S. provided valuable suggestions on phenotypical analysis and interpretation of the results of KO CMs, scRNA-seq and provided critical revision. G.H., F.W., and W.S. supervised the study. All authors reviewed the results and approved the final version of the manuscript.

## Competing interests

The authors declare no competing interests.
