## [Peer Review file · Nature Communications]

Essential role of NONO-HOXA1-Wnt axis in cardiomyocyte differentiation

Corresponding Author: Professor Guoying Huang

Version 0:

Reviewer comments:

Reviewer #1

(Remarks to the Author)

The manuscript by Feng et al. presents a comprehensive investigation into the role of the non-POU domain-containing octamer-binding protein (NONO) in heart development, using human iPSC-directed cardiomyocyte (CM) differentiation as a model system. The authors generated a series of hiPSC lines to address this question. Specifically, they employed CRISPR/Cas9 to compare the CM differentiation in NONO-KO hiPSC to that of WT hiPSC. RNAseq analysis revealed that genes involved in early heart specification were downregulated in NONO-KO hiPSC, resulting in impaired CM maturation in NONO-KO cells, accompanied by structural and functional abnormalities. These defects were rescued upon reactivation of NONO (NONO-RE) in NONO-KO hiPSC-derived CMs. Transcriptomic analysis of WT, NONO-KO and NONO-RE hiPSCs at differentiation days 2, 5, 9 and 15 showed that NONO modulates the expression of genes critical for heart development. Given that NONO regulates both transcription and alternative splicing, the authors further analyzed transcriptomic data at day 2 and compared these findings with NONO-ChIP-seq data. Based on this comparison, they suggest that NONO regulates gene expression through transcriptional activity, independently of its splicing function. To explore how NONO influences gene expression in CMs, the authors performed immunoprecipitation-mass spectrometry and identified HOXA1 as a NONO interactor, binding through its homeodomain. ChIP-seq analysis at day 2 revealed 1,169 overlapping peaks between NONO and HOXA1, with an enrichment in genes involved in processes such as Wnt signaling pathway, gastrulation, and heart morphogenesis. These data led the authors to propose that NONO is indispensable for HOXA1-mediated regulation of precardiac mesoderm genes. Finally, focusing on canonical Wnt signaling, they show that the NONO-HOXA1 interaction is critical for Wnt/beta-Catenin activation.

This study provides a rich dataset combining mRNA-seq and ChIP-seq in hiPSC-derived CMs, offering mechanistic insights into the role of NONO in heart development. The use of bioinformatic analyses enhances our understanding of how NONO regulates CM specification/differentiation. The study identifies NONO-HOXA1 interaction as a key regulator in cardiac development. However, a major limitation of the study is the lack of experimental validation for key conclusions, particularly the proposed role of the NONO-HOXA1 interaction in regulating early cardiac genes. The manuscript would also benefit from a clearer discussion of how these results advance beyond previous works on the role of NONO during early development.

Overall, the study is thoughtfully conducted and offers valuable observations that contribute to our understanding of early heart development. Nonetheless, the central conclusion regarding the NONO-HOXA1-Wnt signaling axis in cardiac specification is not sufficiently substantiated and would be strengthened by additional experimental validation.

Major points:

- The authors state that NONO plays a critical role in CM differentiation. However, based on the presented data, it appears more accurate to conclude that NONO is required for early specification, and that disruption at this stage subsequently impairs CM differentiation. To clarify this point, it is important to quantify the proportion of progenitor cells expressing MESP1, GATA4, and NKX2-5. It remains unclear whether the NONO-KO cells fail to specify into cardiac progenitors or adopt non-cardiac lineages instead.
- The authors should specify which subtype of CMs are generated using their differentiation protocol. It is well established that in vitro differentiation of hiPSC can give rise to heterogeneous CM populations. Clarifying the major type of CM subtype would help also to understand the phenotype during CM maturation.
- The authors demonstrate that reactivation of NONO expression in NONO-KO hiPSCs rescues CM differentiation. However, to fully support their model, it is essential to also show that early cardiac specification is restored upon this reactivation.

- NONO expression peaks at day 2, which is consistent with a potential role in early cardiac specification. To support this, the authors should indicate the timing of CM differentiation by presenting the temporal expression profile of key differentiation markers.
- The comparison of gene expression levels between NONO-RE and NONO-KO cells from day 2 to day 15 identified 332 genes in Group I and 60 genes in Group I', which are potentially regulated by NONO. However, it remains unclear whether these genes are direct or indirect targets of NONO. To address this point, the authors should integrate and compare the NONO ChIP-seq data with these gene sets to determine which are directly bound by NONO. This analysis would help distinguish primary regulator targets from downstream effects.
- The authors should address the temporal differences between their in vitro differentiation model and in vivo cardiac development during embryogenesis. Notably, several studies have established that MESP1 as a critical factor for early cardiac specification, acting upstream of key transcription factors in both the FHF and SHF (e.g., Lescroart et al. 2014, 2018). In this context, it is unexpected that MESP1 expression peaks at day 4 in the present model, as MESP1 expression in vivo occurs much earlier during mesodermal specification. This point should be more detailed and included in the discussion.
- The analysis of DEG and ASE between WT and NONO-KO cells, in conjunction of NONO-targeted genes, identified 542 transcriptionally regulated genes enriched in heart development pathways. However, as shown in Supplementary Fig 4C, there is an additional category of genes involved in the negative regulation of stem cell differentiation, which is also highly relevant and should be discussed. Pointing to this category would provide a more comprehensive view of how NONO may influence both lineage commitment and the suppression of alternative fates during early cardiac differentiation.
- The immunoprecipitation-mass spectrometry analysis performed at day 2 identified NONO and HOXA1 as potential interacting partners. The authors should clarify whether NONO forms homodimers, as this may influence its mode of interaction with other proteins. Additionally, since HOXA1 is known to form homodimers, it is important to address the possibility of protein competition or cooperation between HOXA1 homodimers and HOXA1-NONO complexes in regulating target gene expression.
- To strengthen the analysis of overlapping HOXA1 and NONO binding peaks, it would be essential to include additional epigenetic data, such as histone modification marks or ATAC-seq profiles. These data would help to identify active enhancer regions.
- Figure 5F: The peak enrichment at the PDGFRA and MESP1 loci shown in Fig. 5F is not convincing. The authors should either provide higher-resolution tracks, quantitative peak calling data or additional replicates to support their claims.

Minor Points:

- Figure 6F: SiCon must be changed to NC.

(Remarks on code availability)

Reviewer #2

(Remarks to the Author)

In this manuscript, the authors showed interesting findings about the molecular mechanism of NONO in cardiomyocyte differentiation. Key contribution of NONO in cardiomyocyte differentiation is confirmed, and it is also demonstrated to be independent of NONO's known splicing function. Furthermore, the authors identify HOXA1 as a key effector of NONO, and establish that the domain HD and residue K286 of HOXA1 play an essential role in the direct interaction. The HOXA1-NONO complex regulates precardiac mesoderm gene expression by active downstream Wnt signaling. This manuscript is written fluently and is convinced by detailed data. However, there are still some major problems to be solved.

Major Concerns:

1. The authors did not properly distinguish the concepts between differentiation and maturation. Maturation means the developmental process from a naïve cardiomyocyte to a better functional cardiomyocyte. The authors differentiated the cells for only 15 days and did not use methods to boost hiPSC-CM maturation (T3, fatty acid, electric stimulation or EHT formation). Therefore, all the conclusions about cardiomyocyte maturation are inappropriate and are indeed about differentiation. Here I recommend the authors to replace all maturation by differentiation and to avoid mentioning maturation, which the authors did not really examine.
2. The cell composition of the NONO-KO versus control hiPSC-CM populations are apparently different due to the impaired differentiation efficiency in the mutant group. Consequently, bulk RNA-seq would largely reflect cell composition changes rather than transcription changes in CMs or CPCs. To address this concern, the authors should add scRNA-seq validation and look at the cardiac progenitor cell population to see if the observed gene expression changes are still real.
3. The authors mentioned that NONO is likely a chaperone, thus the authors are recommended to look at HOXA1 stability to check this point. In addition, although the authors mapped the binding sites in HOXA1, they did not map the protein interaction sites in NONO and determine if this binding is necessary for HOXA1 stability and function.
4. HD domain is responsible for HOXA1-DNA binding. Thus it is critical to test how NONO affects HOXA1 affinity to DNA. In addition, the authors emphasized that HOXA1 ChIP-seq peak decreased in the NONO-KO cells. This could still be due to changes in cell composition rather than NONO-KO.
5. The authors are recommended to validate key findings in an animal model, since the hiPSC-CM differentiation process is still quite artificial and may not truly reflect in vivo development.
6. The mechanism how HOXA1 and NONO are required for nuclear localization of β -catenin lacks further explanation. If the cytoplasmic β -catenin level does not change, how could it be a nuclear translocation phenotype? Alternatively, could these factors affect nuclear β -actinin stability?
7. The authors used a classic two-small molecule differentiation method that involves WNT activation. If NONO-KO truly

impair WNT activation and thereby CM differentiation, then one would expect rescued cardiomyocyte differentiation simply by increasing the concentration of CHIR activating Wnt signaling at the first step of differentiation? This might be a stronger support for Wnt signaling mechanism.

8. NONO genetic variants are found in CHD patients. The results in this study should be validated in a hiPSC line with patient-related genetic variants to make sure the results in this study is disease related.

9. The conclusion that NONO cooperates with HOXA1 is weak. The authors could do a reChIP experiment to see if the two molecules indeed act on the same loci. In addition, NONO-HOXA1 double silencing cells should be compared to each single KO lines to show cooperativity, rather than a upstream-downstream relationship in the same pathway.

Minor Comments:

1. A karyotype detection of the new hiPSC lines should be supplied. It is ideal to test the results in multiple hiPSC clones, even though the authors claimed that they are similar.
2. Fig 1F: the images do not show reduced peak intensity, but rather increased.
3. Fig2A: in the rescue experiment, the restored NONO protein level should be compared to endogenous level to determine why there is only partial rescue.
4. Fig3B should label clearly that the genes were all from group I. In addition, the authors should display how the rescue changes genes in each group, not just group I.
5. In line 569 and 618, the word "hESC" is misused. Human embryonic stem cell (hESC) is different from human induced pluripotent stem cell (hiPSC). The authors should be clear what cells are used for each cell line.

(Remarks on code availability)

Version 1:

Reviewer comments:

Reviewer #1

(Remarks to the Author)

In this revised manuscript, the authors have thoroughly addressed all my comments. The overall quality of the study has been subsequently improved, and the data are now clearly presented and well interpreted.

In response to my comments, the authors have refined their conclusions regarding the role of NONO in early cardiac specification rather than late-stage maturation. Additional experiments, including new flow cytometry and single-cell RNA-seq analyses to quantify cardiac progenitor populations and cell-state transitions, have helped them better assess the role of NONO during precardiac mesoderm development. Cardiomyocytes (CM) are now better characterized using markers such as Myl2 and Myl7, which indicate that the CMs generated have predominantly atrial identity. The rescue experiments are also better characterized thanks to the inclusion of early markers. A temporal expression analysis, shown in Supplementary Figure 2, highlights the progression of CM differentiation in their model. Integration of the NONO ChIP-seq data with transcriptomic profiles has enabled the authors to identify direct targets of NONO. Clarifying the temporal relationships between NONO, MESP1, and key developmental markers has strengthened the understanding of their model and the role of NONO in CM differentiation. The discussion now includes a section on the "negative regulation of stem cell differentiation" gene set and provides a more careful comparison of their findings with in vivo and previous studies. Addressing the homodimerization of NONO and HOXA1 has further strengthened the mechanistic model of NONO-HOXA1 cooperation by characterizing their dimerization properties. Through sophisticated approaches such as Re-ChIP, they map interaction domains and assess how loss of NONO affects HOXA1 chromatin binding and downstream Wnt signaling.

Overall, the revised manuscript demonstrates significant improvements in clarity, and mechanistic insight.

(Remarks on code availability)

The GitHub link provides access to the bioinformatic datasets and analysis scripts used in this study.

Reviewer #2

(Remarks to the Author)

The authors did a commendable work at addressing my comments. Most of my questions were well answered by supplemental experiments, including scRNA-seq (for the major comments 2), protein stability assays (for the major comments 3 and 6), domain-mapping experiments (for the major comments 3), mechanistic analyses (for the major comments 4), CHIR99021 dose-response experiment (for the major comments 7), validation in hiPSC lines with patient-related genetic variants (for the major comments 8), reChIP (for the major comments 9), double knockout hiPSC lines generation (for the major comments 9) and karyotype detection (for the minor comments 1). For the major comments 5, although the authors didn't provide additional experiments, they gave a reasonable explanation in the discussion. The authors also corrected their misuse of "maturation" and "hiPSC". The only imperfection is that replicates and a quantification are needed for Supplementary Fig. 4B. I have no further requests.

(Remarks on code availability)

Response to Reviewers

We sincerely thank both Reviewer #1 and Reviewer #2 for their thoughtful and constructive comments. Their insightful suggestions have significantly improved the clarity, rigor, and overall quality of our manuscript. Below, we provide a point-by-point response to each comment. Reviewer comments are shown in *italic*. All corresponding revisions have been incorporated into the main text, Materials and Methods, Supplementary Information, and figures.

Reviewer #1 (Remarks to the Author):

The manuscript by Feng et al. presents a comprehensive investigation into the role of the non-POU domain-containing octamer-binding protein (NONO) in heart development, using human iPSC-directed cardiomyocyte (CM) differentiation as a model system. The authors generated a series of hiPSC lines to address this question. Specifically, they employed CRISPR/Cas9 to compare the CM differentiation in NONO-KO hiPSC to that of WT hiPSC. RNAseq analysis revealed that genes involved in early heart specification were downregulated in NONO-KO hiPSC, resulting in impaired CM maturation in NONO-KO cells, accompanied by structural and functional abnormalities. These defects were rescued upon reactivation of NONO (NONO-RE) in NONO-KO hiPSC-derived CMs. Transcriptomic analysis of WT, NONO-KO and NONO-RE hiPSCs at differentiation days 2,5,9 and 15 showed that NONO modulates the expression of genes critical for heart development. Given that NONO regulates both transcription and alternative splicing, the authors further analyzed transcriptomic data at day 2 and compared these findings with NONO-ChIP-seq data. Based on this comparison, they suggest that NONO regulates gene expression through transcriptional activity, independently of its splicing function. To explore how NONO influences gene expression in CMs, the authors performed immunoprecipitation-mass spectrometry and identified HOXA1 as a NONO interactor, binding through its homeodomain. ChIP-seq analysis at day 2 revealed 1,169 overlapping peaks between NONO and HOXA1, with an enrichment in genes involved in processes such as Wnt signaling pathway, gastrulation, and heart morphogenesis. These data led the authors to propose that NONO is indispensable for HOXA1-mediated regulation of precardiac mesoderm genes. Finally, focusing on canonical Wnt signaling, they show that the NONO-HOXA1 interaction is critical for Wnt/beta-Catenin activation.

This study provides a rich dataset combining mRNA-seq and ChIP-seq in hiPSC-derived CMs, offering mechanistic insights into the role of NONO in heart development. The use of bioinformatic analyses enhances our understanding of how NONO regulates CM specification/differentiation. The study identifies NONO-HOXA1 interaction as a key regulator in cardiac development. However, a major limitation of the study is the lack of experimental validation for key conclusions, particularly the proposed role of the NONO-HOXA1 interaction in regulating early cardiac genes. The manuscript would also benefit from a clearer discussion of how these results advance beyond previous works on the role of NONO during early development.

Overall, the study is thoughtfully conducted and offers valuable observations that contribute to our understanding of early heart development. Nonetheless, the central conclusion regarding the NONO-HOXA1-Wnt signaling axis in cardiac specification is not sufficiently substantiated and would be strengthened by additional experimental validation.

We thank the reviewer for the constructive assessment and for recognizing the value of our genetic models and the proposed NONO–HOXA1–Wnt mechanistic framework. This feedback has significantly improved the clarity and rigor of our manuscript.

We have carefully addressed the reviewer's suggestions regarding experimental validation and the differentiation of our work from prior studies. Accordingly, we have incorporated new data to bolster our conclusions and expanded the Discussion to better define the novelty of our findings in the context of early cardiac development. A point-by-point response detailing these revisions follows below.

Reviewer #1 – Major points 1:

- The author state that NONO plays a critical role in CM differentiation. However, based on the presented data, it appears more accurate to conclude that NONO is required for early specification, and that disruption at this stage subsequently impairs CM differentiation. To clarify this point, it is important to quantify the proportion of progenitor cells expressing MESP1, GATA4, and NKX2-5. It remains unclear whether the NONO-KO cells fail to specify into cardiac progenitors or adopt non-cardiac lineages instead.

Response:

We thank the reviewer for this important clarification. We agree that our data more accurately support the conclusion that NONO is required for early cardiac specification, and that impaired cardiomyocyte differentiation in NONO-KO cells arises as a downstream consequence of defective lineage induction.

To directly address the reviewer's concern, we quantified the proportions of early cardiac progenitor populations at key differentiation stages. Because MESP1 is not reliably detected by flow cytometry, we used PDGFRA surface expression at day 3 as an established proxy for MESP1⁺ mesodermal progenitors¹. Consistent with defective specification, NONO-KO cells showed a marked reduction in PDGFRA⁺ cells compared with WT (67.0% vs. 83.5%). This reduction persisted at

A-C Flow cytometry analysis showing reduced proportions of PDGFRA-positive cells at Day 3, GATA4-positive cells at Day 5, and NKX2-5-positive cells at Day 6. $n = 3$. P values were calculated using two-tailed unpaired Student's t-test. $P < 0.05$ was considered significant. **D** UMAP plots of WT and NONO-KO cells at Days 2, 5, and 15, illustrating the major cell populations during differentiation. **E** Violin plots showing the expression levels of key cardiac lineage regulators (*MESP1*, *ISL1*, *GATA4*, *HAND1*, *TBX20*, *POU5F1*) and *NONO* within CPC populations of WT and NONO-KO cells at Day 5 of differentiation.

subsequent specification checkpoints, with fewer GATA4⁺ progenitors at day 5 (18.3% vs. 35.0%) and fewer NKX2-5⁺ cardiac progenitors at day 6 (41.2% vs. 77.2%) (Panels A–C).

Our newly added single-cell RNA-seq analysis further supports this interpretation. Although the CPC-like cluster is present in both genotypes, NONO-KO progenitors exhibit significantly reduced expression of core cardiac transcription factors such as GATA4 and ISL1, and an increased bias toward neural-associated lineages at day 5 (Panels D–E). These data demonstrate that NONO-KO cells do not undergo a complete loss of mesoderm formation but instead fail to properly activate the cardiac specification program, leading to diminished downstream differentiation potential.

Together, these results clarify that NONO functions at the specification stage and that mis-specification, rather than an intrinsic block in cardiomyocyte differentiation, underlies the observed differentiation defects.

The results shown here in Panels A–C have been integrated into the revised manuscript as **Supplementary Fig. 2E–J**. Similarly, the data in Panel D and Panel E are now presented as **Supplementary Fig. 3D and Fig. 1L**, respectively. Corresponding descriptions have been updated in the revised text (**Page 4, lines 133–136; Page 6, lines 197–200**).

Reviewer #1 - Major points 2:

- The authors should specify which subtype of CMs are generated using their differentiation protocol. It is well established that in vitro differentiation of hiPSC can give rise to heterogeneous CM populations. Clarifying the major type of CM subtype would help also to understand the phenotype during CM maturation.

Response:

We thank the reviewer for this insightful comment. We agree that defining the cardiomyocyte (CM) subtype composition is important for interpreting the differentiation phenotype observed in our system. To address this point, we analyzed our day 15 single-cell RNA-seq dataset and examined the expression of canonical atrial and ventricular markers within TNNT2⁺ cardiomyocytes.

This analysis showed that TNNT2⁺ cells exhibit predominant MYL7 expression (73.9%) and minimal MYL2 expression (0.4%) (Panel A), indicating that the majority of CMs generated at this stage resemble atrial-like or immature ventricular-like cardiomyocytes. This is consistent with the known developmental timeline in hiPSC-derived cardiomyocytes, where MYL7 (MLC2a) is expressed early, whereas MYL2 (MLC2v)—a hallmark of ventricular maturation—typically appears only at substantially later stages of in vitro differentiation (often > day 40).

^{2,3}

Together, these data clarify that our differentiation protocol predominantly produces immature, MYL7-enriched cardiomyocytes at day 15, which is consistent with the early-stage differentiation phenotypes characterized in our study.

A Single-cell RNA-seq analysis showing expression of the ventricular marker MYL2 and the early/atrial-enriched marker MYL7 in TNNT2⁺ cardiomyocytes at day 15.

The results shown here in Panels A have been integrated into the revised manuscript as **Supplementary Fig. 2C**. Corresponding descriptions have been updated in the revised text (**Page 5, lines 126–129**).

Reviewer #1 - Major points 3:

- *The authors demonstrate that reactivation of NONO expression in NONO-KO hiPSCs rescues CM differentiation. However, to fully support their model, it is essential to also show that early cardiac specification is restored upon this reactivation.*

Response:

We thank the reviewer for this important comment. We fully agree that demonstrating the rescue of early cardiac specification is essential to support our model that NONO acts upstream of lineage commitment and that its reactivation functionally restores the differentiation trajectory.

To directly address this point, we examined the temporal expression of key mesodermal and early cardiac transcription factors from day 2 to day 7 in NONO-RE cells. Quantitative RT-PCR analysis showed that NONO reactivation robustly restored the induction of *T*, *EOMES*, *MIXL1*, *MESP1* and *PDGFRA* at early mesoderm stages, and subsequently rescued the activation of *GATA4*, *NKX2-5* and *MYOCD* during cardiac progenitor formation (Panels A–B). The restored expression levels closely mirrored those observed in WT cells.

These results demonstrate that NONO reactivation effectively reinstates the early cardiac specification program, thereby providing functional evidence that the impaired differentiation phenotype in NONO-KO cells arises from defective lineage induction rather than an inability to undergo later maturation. This rescue further strengthens our conclusion that NONO is required at the specification stage to initiate the transcriptional cascade necessary for proper cardiomyocyte development.

A, B Expression levels of cardiac lineage markers at Day 2 (*T*, *EOMES*, *MIXL1*, *MESP1*, and *PDGFRA*), Day 3 (*MIXL1*, *MESP1*, and *PDGFRA*), Day 5 (*GATA4*), Day 6 (*GATA4*, *NKX2-5*, and *MYOCD*), and Day 7 (*NKX2-5* and *MYOCD*) in NONO-KO^{NONO OE} (with or without Dox) cells during differentiation. P values were calculated using two-tailed unpaired Student's t-test. P < 0.05 was considered significant.

The results shown here in Panels A, B have been integrated into the revised manuscript as **Fig. 2B, C**. Corresponding descriptions have been updated in the revised text (**Page 9, lines 249–253**).

Reviewer #1 - Major points 4:

- *NONO* expression peaks at day 2, which is consistent with a potential role in early cardiac specification. To support this, the authors should indicate the timing if CM differentiation by presenting the temporal expression profile of key differentiation markers.

Response:

We thank the reviewer for this insightful suggestion. We agree that defining the temporal dynamics of NONO relative to established lineage markers is essential for clarifying its developmental role.

To address this point, we performed a detailed RT-qPCR time-course analysis across cardiac differentiation (days 0, 1, 2, 3, 4, 5, 6, 7, 9, and 15), examining the expression trajectories of pluripotency, mesoderm, cardiac progenitor, and cardiomyocyte markers, including *OCT4*, *T*, *EOMES*, *MESP1*, *ISL1*, *GATA4*, *PDGFRA*, *NKX2-5*, *MYH6*, *MYH7*, and *TNNT2* (Panel A).

This analysis revealed that NONO expression is transiently upregulated at day 2, coinciding with the induction of early mesodermal regulators T and EOMES, and immediately preceding the peak of the cardiac mesoderm marker MESP1 at day 3. The subsequent activation of ISL1, GATA4, and NKX2-5 further delineates the normal progression toward cardiac progenitor formation.

The alignment of the NONO peak with these early lineage transitions strongly supports the conclusion that NONO acts at a pre-specification to early-specification window, functioning upstream of the transcriptional network that initiates cardiac lineage commitment.

A Temporal expression profiles of the pluripotency marker *OCT4*, mesoderm markers (*T* and *EOMES*), cardiac progenitor genes (*MESP1*, *ISL1*, *GATA4*, and *PDGFRA*), and cardiomyocyte genes (*NKX2-5*, *TNNT2*, *MYH6*, and *MYH7*) during hiPSC differentiation, quantified by RT-qPCR. n=3.

The results shown here in Panels A have been integrated into the revised manuscript as **Supplementary Fig. 2B**. Corresponding descriptions have been updated in the revised text (**Page 4 and 5, lines 114–126**).

Reviewer #1 - Major points 5:

- *The comparison of gene expression levels between NONO-RE and NONO-KO*

cells from day 2 to day 15 identified 332 genes in Group I and 60 genes in Group I', which are potentially regulated by NONO. However, it remains unclear whether these genes are direct or indirect targets of NONO. To address this point, the authors should integrate and compare the NONO ChIP-seq data with these gene sets to determine which are directly bound by NONO. This analysis would help distinguish primary regulator targets from downstream effects.

Response:

Thank you for this important suggestion. In response, we performed an integrative analysis combining our NONO ChIP-seq dataset with the 332 Group I genes and 60 Group I' genes identified from the NONO-RE versus NONO-KO comparisons. Using the day 2 NONO ChIP-seq dataset, we identified 4,350 NONO peaks associated with 2,212 gene promoters. Intersecting these promoter-associated peaks with our DEG groups revealed that 23 of the 332 Group I genes and 3 of the 60 Group I' genes exhibit direct NONO promoter binding at day 2. These genes therefore represent the most likely primary transcriptional targets of NONO.

We note, however, that the expression differences defining Group I and I' genes were derived from comparisons spanning day 2 to day 15, a developmental interval during which transcriptional regulatory networks undergo substantial and stage-specific rewiring. Because the available NONO ChIP-seq data capture binding only at day 2, many DEGs emerging at later stages are expected to be regulated indirectly, through downstream transcription factors, signaling changes, or lineage-specific cascades initiated earlier in differentiation.

Together, these integrative analyses distinguish a small subset of genes with direct NONO promoter occupancy from a broader set of indirectly regulated genes, thereby providing a clearer interpretation of NONO's primary versus downstream regulatory effects.

Reviewer #1 - Major points 6:

- *The authors should address the temporal differences between their in vitro differentiation model and in vivo cardiac development during embryogenesis. Notably, several studies have established that MESP1 as a critical factor for early cardiac specification, acting upstream of key transcription factors in both the FHF and SHF (e.g., Lescroart et al. 2014, 2018). In this context, it is unexpected that MESP1 expression peaks at day 4 in the present model, as MESP1 expression in vivo occurs much earlier during mesodermal specification. This point should be more detailed and included in the discussion.*

Response:

We thank the reviewer for this insightful comment. We apologize for the confusion caused by the missing day-3 label in the initial figure, which may have led to the impression that MESP1 peaks at day 4. After correcting the X-axis and presenting the full time course, our data clearly show that MESP1 is induced at day 2 and peaks at day 3, which is consistent with previously reported PSC-based cardiac differentiation systems³ (as also addressed in our response to Major points 4). We agree with the reviewer that timing differences between *in vitro* hiPSC differentiation and *in vivo* embryogenesis are well recognized. We have now expanded the Discussion to acknowledge these temporal differences and to clarify how our results align with established *in vivo* findings. We also note that future studies using mouse embryonic models will be necessary to examine the precise *in vivo* temporal relationship between NONO, MESP1, and early mesodermal specification.

Discussion part:

Regarding the temporal dynamics of this regulation, we noted that *MESP1* induction peaks at day 3 in our hiPSC differentiation model (Supplementary Fig. 2B). While this timing is slightly delayed compared to mesodermal specification *in vivo*, it is consistent with established kinetics in pluripotent stem cell-based differentiation systems⁴⁻⁶. We acknowledge that *in vitro* models cannot fully recapitulate the precise chronometry of embryonic development, nor do they possess the complex spatial architecture of native organogenesis. However, the hiPSC system offers an ideal, controlled platform for dissecting molecular mechanisms. Therefore, validating our findings *in vivo* is a critical next step. Future studies using *Nono*-deficient mice will determine whether the molecular defects observed here translate into anatomical malformations during heart development.

Corresponding descriptions have been updated in the revised text (Page 28, lines 1084–1094).

Reviewer #1 - Major points 7:

- *The analysis of DEG and ASE between WT and NONO-KO cells, in conjunction of NONO-targeted genes, identified 542 transcriptionally regulated genes enriched in heart development pathways. However, as shown in Supplementary Fig 4C, there is an additional category of genes involved in the negative regulation of stem cell differentiation, which is also highly relevant and should be discussed. Pointing to this category would provide a more comprehensive view of how NONO may influence both lineage commitment and the suppression of alternative fates during early cardiac differentiation.*

Response:

We appreciate the reviewer for highlighting the biological significance of the "negative regulation of stem cell differentiation" category. We agree that discussing this category provides a more comprehensive view of NONO's role in balancing lineage commitment and the suppression of alternative fates.

In the revised Discussion, we have added a dedicated section to address this. We now explain that NONO likely functions to dampen the expression of these "gatekeeper" genes (such as *EZH2*, *REST*, and *HES1*). This discussion clarifies that NONO is required not only to activate cardiac pathways but also to overcome the molecular barriers that restrict differentiation, thereby ensuring a robust exit from the progenitor state.

Heatmap of “negative regulation of stem cell differentiation” genes in WT and NONO-KO cells at day 2. Scaled expression values (0–1) of key progenitor-maintenance regulators (*EZH2*, *PRICKLE1*, *REST*, *HSPA9*, *YTHDF2*, *HES1*, *NFE2L2*) show marked upregulation in NONO-KO samples compared to WT controls.

Discussion part:

Beyond activating cardiac developmental pathways, our analysis revealed that NONO actively suppresses lineage-restricting factors. Established repressors such as *EZH2*⁷, *REST*⁸, and *HES1*⁹ were aberrantly upregulated in NONO-KO cells (Supplementary Fig 7D). This suggests a dual mechanism wherein NONO facilitates lineage commitment by synergistically activating cardiac identity genes while simultaneously dismantling transcriptional barriers imposed by progenitor-maintenance programs.

The results shown here in Panel have been integrated into the revised manuscript as **Supplementary Fig. 7D**. Corresponding descriptions have been updated in the revised text (**Page 28, lines 1079–1084**).

Reviewer #1 - Major points 8:

- The immunoprecipitation-mass spectrometry analysis performed at day 2 identified NONO and HOXA1 as potential interacting partners. The authors should clarify whether NONO forms homodimers, as this may influence its mode of interaction with other proteins. Additionally, since HOXA1 is known to form homodimers, it is important to address the possibility of protein competition or cooperation between HOXA1 homodimers and HOXA1-NONO complexes in regulating target gene expression.

Response:

We thank the reviewer for these insightful comments, which have significantly strengthened our mechanistic understanding of the HOXA1–NONO interaction. In particular, the questions regarding the interplay between HOXA1 homodimers and HOXA1–NONO complexes prompted us to further examine how NONO affects HOXA1 dimerization and DNA binding.

In response, we performed additional experiments to directly address the dimerization properties of NONO and HOXA1 and the potential regulatory interplay between HOXA1 homodimers and HOXA1–NONO complexes.

1. NONO homodimerization

To experimentally validate whether NONO forms homodimers in our differentiation system, we generated doxycycline-inducible NONO-FLAG and NONO-HA hiPSC lines. Doxycycline was added at day 1 of differentiation, and co-expression of NONO-FLAG and NONO-HA was induced.

Consistent with the reviewer's comment, co-immunoprecipitation (co-IP) experiments demonstrated that NONO forms homodimers in day-2 differentiated cells (Panel A). This property is likely to influence its mode of interaction with other transcriptional regulators, including HOXA1.

2. Competition or cooperation between HOXA1 homodimers and HOXA1-NONO complexes

To further investigate whether NONO influences HOXA1 dimerization, we established doxycycline-inducible HOXA1-FLAG and HOXA1-HA hiPSC lines in both WT and NONO-KO backgrounds. Doxycycline was added at day 1 of differentiation, and HOXA1 overexpression was induced.

The co-IP analyses revealed that HOXA1 homodimer formation was markedly reduced in NONO-KO cells compared with WT controls (Panel B and C). Based on our current data, we favor a cooperative rather than competitive relationship between the NONO–HOXA1 complex and HOXA1 homodimers.

A Co-immunoprecipitation (Co-IP) of Day-2 hiPSC-differentiated cells with doxycycline-induced NONO-FLAG and NONO-HA expression, using anti-FLAG and anti-HA antibodies for immunoprecipitation and detection. **B, C** Co-immunoprecipitation (Co-IP) of WT and *NONO*-KO Day-2 hiPSC-differentiated cells with doxycycline-induced HOXA1-HA and HOXA1-FLAG expression, using anti-HA (**B**) and anti-FLAG (**C**) antibodies for immunoprecipitation, followed by immunoblotting with anti-FLAG and anti-HA antibodies.

The results shown here in Panel A-C have been integrated into the revised manuscript as **Fig. 5J-L**. Corresponding descriptions have been updated in the revised text (**Page 20, lines 776–783**).

Reviewer #1 - Major points 9:

- To strengthen the analysis of overlapping *HOXA1* and *NONO* binding peaks, it would be essential to include additional epigenetic data, such as histone modification marks or ATACseq profiles. These data would help to identify active enhancer regions.

Response:

We thank the reviewer for this constructive suggestion. To characterize the epigenetic landscape of the *NONO*-*HOXA1* co-occupied regions, we performed ATAC-seq and H3K27ac ChIP-seq analyses in Day-2 differentiated cells. By intersecting these profiles with our *NONO* and *HOXA1* ChIP-seq data, we found that the vast majority of *NONO*-*HOXA1* co-occupied peaks reside within accessible chromatin regions (~91% overlap with ATAC-seq peaks, $1110 / (1110+104) \times 100 = 91\%$) and are marked by active histone modifications (~72% overlap with H3K27ac peaks, $901 / (901+349) \times 100 = 72\%$) (Panel A). these data

support a model where NONO and HOXA1 cooperate at open, active chromatin regions to regulate target gene expression.

A NONO and HOXA1 co-occupy accessible and active chromatin regions.

The processed peak calling data are available at:

<https://drive.google.com/drive/folders/1wzqwnzLNZB0LX3Ik0ojFa1qiOeZmrRnf?usp=sharing>

Reviewer #1 - Major points 10:

- Figure 5F: The peak enrichment at the PDGFRA and MESP1 loci shown in Fig. 5F is not convincing. The authors should either provide higher-resolution tracks, quantitative peak calling data or additional replicates to support their claims.

Response:

We appreciate the reviewer's rigorous scrutiny regarding the ChIP-seq enrichment. To provide more quantitative support beyond the browser tracks, we performed sequential ChIP (Re-ChIP) assays using a doxycycline-inducible HOXA1-HA hiPSC line differentiated to day 2. Chromatin was first immunoprecipitated with an anti-HA antibody (targeting HOXA1), eluted under native conditions, and subsequently re-immunoprecipitated with an anti-NONO antibody. As shown in Panel A, we observed robust enrichment at the MESP1 locus in the primary HA-ChIP, and PDGFRA demonstrated significant enrichment in the NONO Re-ChIP fraction, indicating that both loci are co-occupied by HOXA1 and NONO. Together with the updated higher-resolution browser tracks, these Re-ChIP results provide strong and direct evidence supporting NONO–HOXA1 co-occupancy at key precardiac mesoderm regulatory loci, including MESP1 and PDGFRA.

A ChIP-seq browser tracks for representative target precardiac mesoderm genes (MESP1 and PDGFRA) showing HOXA1 binding in day 2 differentiated WT cells, with reduced binding in NONO-KO cells.

Minor Points:

The results shown here in Panel A have been integrated into the revised manuscript as **Fig. 5F**. Corresponding descriptions have been updated in the revised text (**Page 19 and 20, lines 730–761**).

Reviewer #1 -Minor point:

-Figure 6F: SiCon must be changed to NC.

Response:

We thank the reviewer for noting this oversight. Fig.6 “SiCon” has been corrected to “NC” as suggested.

Summary and final appreciation for Reviewer #1

We are deeply grateful to Reviewer #1 for the thorough and constructive evaluation of our work. In response to these comments, we have (i) refined our conceptual framework to emphasize NONO’s role in early cardiac specification rather than late-stage maturation, (ii) added new flow cytometry and single-cell RNA-seq analyses to quantify cardiac progenitor populations and cell-state transitions, (iii) integrated NONO ChIP-seq data with time-resolved transcriptomic profiles to distinguish direct from indirect targets, (iv) clarified the temporal relationship between NONO, MESP1, and key developmental markers, and (v) expanded the Discussion to incorporate the “negative regulation of stem cell differentiation” gene set and to more carefully compare our findings with in vivo and previous studies. In addition, we have strengthened the mechanistic model of NONO–HOXA1 cooperation by characterizing their dimerization properties, mapping interaction domains, performing Re-ChIP and epigenomic profiling (ATAC-seq and H3K27ac ChIP-seq), and examining the impact of NONO loss on HOXA1 chromatin binding and downstream Wnt signaling. Together, these revisions have, in our view, substantially improved the clarity, depth, and robustness of the manuscript. We sincerely thank Reviewer #1 again for the insightful guidance that has been instrumental in shaping the revised version of this study.

Reviewer #2 (Remarks to the Author):

In this manuscript, the authors showed interesting findings about molecular mechanism of NONO in cardiomyocyte differentiation. Key contribution of NONO in cardiomyocyte differentiation is confirmed, and it is also demonstrated to be independent of NONO's known splicing function. Furthermore, the authors identify HOXA1 as a key effector of NONO, and establish that the domain HD and residue K286 of HOXA1 play essential role in the direct interaction. The HOXA1-NONO complex regulates precardiac mesoderm gene expression by active downstream Wnt signaling. This manuscript is written fluently and is convinced by detailed data. However, there are still some major problems to be solved.

We thank Reviewer #2 for the positive overall evaluation of our study and for recognizing the mechanistic insights we provide into NONO function during cardiomyocyte differentiation, including its interaction with HOXA1 and downstream effects on Wnt signaling. We also appreciate the reviewer's careful reading of the manuscript and the constructive identification of several important issues that require further clarification or experimental support. In the responses that follow, we address each of these points in detail and describe the revisions, additional analyses, and new experiments incorporated into the revised manuscript.

Reviewer #2 – Major concerns 1:

Major Concerns:

1. The authors did not properly distinguish the concepts between differentiation and maturation. Maturation means the developmental process from a naïve cardiomyocyte to a better functional cardiomyocyte. The authors differentiated the cells for only 15 days and did not use methods to boost hiPSC-CM maturation (T3, fatty acid, electric stimulation or EHT formation). Therefore, all the conclusions about cardiomyocyte maturation are inappropriate and are indeed about differentiation. Here I recommend the authors to replace all maturation by differentiation and to avoid mentioning maturation, which the authors did not really examined.

Response:

We thank the reviewer for this critical distinction. We agree that given our experimental endpoint of 15 days and the absence of specific maturation stimuli (such as T3, fatty acid supplementation, or electromechanical stimulation), our study reflects the process of cardiac differentiation and the generation of naïve cardiomyocytes, rather than functional maturation.

Following the reviewer's recommendation, we have revised the manuscript to replace the term "maturation" with "differentiation" throughout the text, figure legends, and title.

Reviewer #2 - Major concerns 2:

2. The cell composition of the NONO-KO versus control hiPSC-CM populations are apparently different due to the impaired differentiation efficiency in the mutant group. Consequently, bulk RNA-seq would largely reflect cell composition changes rather than transcription changes in CMs or CPCs. To address this concern, the authors should add scRNA-seq validation and look at the cardiac progenitor cell population to see if the observed gene expression changes are still real.

Response:

We thank the reviewer for raising this important concern. To determine whether the transcriptional differences observed in bulk RNA-seq reflect true regulatory changes within cardiac progenitors or cardiomyocytes rather than altered cell-type composition, we performed time-course single-cell RNA-seq on WT and NONO-KO cells collected at days 2, 5, and 15 of differentiation.

UMAP analysis resolved major developmental populations—including mesendoderm, cardiac progenitors (CPCs), cardiomyocytes (CMs), endoderm, epithelial, endothelial, and neural progenitor cells—consistently across samples and defined by canonical lineage markers (Panels A–B). Cells from the three biological replicates were well integrated, indicating minimal batch effects (Panel C).

Monocle²¹⁰ trajectory analysis further demonstrated a continuous progression from mesendoderm to cardiac progenitors and ultimately cardiomyocytes in WT cells. However, NONO-KO cells showed reduced progression along the cardiac trajectory, with a subset retaining mesendoderm-like features at Day 5 and cardiac progenitor markers such as *ISL1*, *GATA4*, *HAND1*, and *TBX20* exhibiting attenuated induction, accompanied by inappropriate persistence of *POU5F1* (Panel D and E).

CytoTRACE¹¹ analysis further confirmed that NONO-KO cells remained transcriptionally less differentiated than WT counterparts within both CPC and CM clusters (Panel F).

Together, these single-cell datasets demonstrate that the transcriptional defects in NONO-KO cells are intrinsic to cardiac progenitor and cardiomyocyte populations, and are not merely a consequence of altered cell composition in bulk RNA-seq.

A UMAP visualization of integrated single-cell RNA-seq data from WT and NONO-KO hiPSC differentiation, showing major cell populations including mesendoderm (MesEnd), neural progenitor cell (NPC), endoderm, epithelial, endothelial, cardiac progenitor cell (CPC), and cardiomyocyte (CM). **B** Bubble plot showing selected marker genes across the cardiomyocyte differentiation scRNA-seq time course. Bubble size represents the percentage of cells expressing each gene, and color indicates mean expression. **C** UMAP plot colored by sample identity (orig.ident), illustrating the distribution of WT and NONO-KO cells across Days 2, 5, and 15 from three biological replicates. **D** Diffusion maps illustrating the developmental trajectories of WT and NONO-KO cells at the indicated time points, with Mesendoderm (MesEnd), Cardiac progenitor cell (CPC), and Cardiomyocyte (CM) lineages shown in distinct colors. **E** Violin plots showing the expression levels of key cardiac lineage regulators (*MESP1*, *ISL1*, *GATA4*, *HAND1*, *TBX20*, *POU5F1*) and *NONO* within CPC populations of WT and NONO-KO cells at Day 5 of differentiation. **F** Violin plots showing the distributions of CytoTRACE scores for WT and NONO-KO cells across the MesEnd, CPC, and CM lineages.

The results shown here in Panels A–C have been integrated into the revised manuscript as **Supplementary Fig. 3A–C**. Similarly, the data in Panel D–F are

now presented as **Fig. 1K-L**, respectively. Corresponding descriptions have been updated in the revised text (**Page 6, lines 185–193; Page 6, lines 197–200**).

Reviewer #2 - Major concerns 3:

3. The authors mentioned that NONO is likely a chaperone, thus the authors are recommended to look at HOXA1 stability to check this point. In addition, although the authors mapped the binding sites in HOXA1, they did not map the protein interaction sites in NONO and determine if this binding is necessary for HOXA1 stability and function.

Response:

We thank the reviewer for this insightful comment. To determine whether NONO functions as a molecular chaperone for HOXA1 and to define the interaction interface between the two proteins, we performed a combination of protein stability assays and domain-mapping experiments.

First, we generated NONO-KO HEK293T cells and conducted cycloheximide (CHX) chase assays using nuclear lysates. HOXA1-HA protein exhibited markedly accelerated decay in NONO-KO cells compared with WT controls (Panels A–B), indicating that NONO is required to maintain HOXA1 protein stability.

To map the NONO regions responsible for binding HOXA1, we constructed deletion mutants lacking RRM1, RRM2, or the coiled-coil domain (Panel C). Co-immunoprecipitation revealed that removal of RRM2 or the coiled-coil region substantially reduced HOXA1 binding (Panel D), identifying these domains as critical for the interaction.

Guided by our structural model of the HOXA1-HD–NONO-Coil interface (Figure 4J), we further introduced a point mutation (E278A) within the coiled-coil region at a predicted contact site. This single-residue substitution dramatically weakened the NONO–HOXA1 interaction (Panel E). Functionally, CHX chase assays demonstrated that the E278A mutant failed to stabilize HOXA1, whereas WT NONO preserved HOXA1 protein levels over time (Panels F, G).

Together, these results show that NONO stabilizes HOXA1 in the nucleus and that this stabilizing function depends on the integrity of NONO's RRM2 and coiled-coil domains. These findings provide strong mechanistic support for the reviewer's suggestion that NONO plays a chaperone-like role in regulating HOXA1 function.

A, B HOXA1-HA stability in nuclear lysates of WT and *NONO*-KO cells overexpressing HOXA1-HA was assessed by CHX chase assay. $n=3$. Protein levels were normalized to LAMIN A/C and expressed relative to the 0-h time point. **C** Schematic of NONO domain mutants: Δ RRM1 (RNA recognition motif domain 1 deletion), Δ RRM2 (RNA recognition motif domain 2 deletion), Δ Coil (Coiled-coil deletion). **D** Co-IP of HEK293T cells overexpressing HOXA1-HA and truncated NONO using anti-HA for pull-down and anti-FLAG for detection shows diminished interaction upon RRM2 and Coiled coil deletion. **E** Co-IP of HEK293T cells overexpressing GFP-HOXA1-HD and NONO mutants using anti-GFP for pull-down and anti-FLAG for detection shows diminished interaction with the E278A mutant. **F, G** HOXA1-HA stability in nuclear lysates from HEK293T cells co-expressing HOXA1-HA and either WT or E278A NONO-FLAG was assessed by a CHX chase assay. $n=3$. Protein levels were normalized to LAMIN A/C and expressed relative to the 0-h time point.

The results shown here in Panels A–G have been integrated into the revised manuscript as **Fig. 4**. Corresponding descriptions have been updated in the revised text (**Page 15 and 16, lines 549-609**).

Reviewer #2 - Major concerns 4:

4. HD domain is responsible for HOXA1-DNA binding. Thus it is critical to test how NONO affects HOXA1 affinity to DNA. In addition, the authors emphasized that HOXA1 ChIP seq peak decreased in the NONO-KO cells. This could still be due to changes in cell composition rather than NONO-KO.

Response:

We thank the reviewer for emphasizing the importance of evaluating HOXA1–DNA binding, particularly given that the HOXA1 homeodomain (HD) mediates its interaction with chromatin. To clarify whether NONO directly affects HOXA1’s chromatin affinity, we performed additional mechanistic analyses.

First, we examined whether NONO influences HOXA1 homodimerization, a property known to enhance HOXA1 DNA-binding capacity. Co-immunoprecipitation in day-2 differentiated hiPSCs expressing HOXA1-HA and HOXA1-FLAG revealed that HOXA1 homodimer formation is markedly reduced in NONO-KO cells (Panels A–B). We further showed that HOXA1 nuclear protein stability is substantially decreased in NONO-KO cells (Panels C–D). Together, these findings support a model in which NONO promotes HOXA1 chromatin engagement by stabilizing the protein and supporting its dimeric state.

A, B Co-immunoprecipitation (Co-IP) of WT and *NONO*-KO Day-2 hiPSC-differentiated cells with doxycycline-induced HOXA1-HA and HOXA1-FLAG expression, using anti-HA (**A**) and anti-FLAG (**B**) antibodies for immunoprecipitation, followed by immunoblotting with anti-FLAG and anti-HA antibodies. **C, D** HOXA1-HA stability in nuclear lysates of WT and *NONO*-KO cells overexpressing HOXA1-HA was assessed by CHX chase assay. $n=3$. Protein levels were normalized to LAMIN A/C and expressed relative to the 0-h time point.

The results shown here in Panel A, B have been integrated into the revised manuscript as **Fig. 5K, L**. Panel C, D have been integrated into the revised

manuscript as **Fig. 4M**. Corresponding descriptions have been updated in the revised text (**Page 20, lines 779-783; Page 16, lines 598-602**).

Regarding the reviewer's concern that reduced HOXA1 ChIP-seq enrichment in NONO-KO hiPSCs might be attributable to altered cell composition rather than loss of NONO itself, we performed an additional experiment in a controlled, lineage-independent system. HOXA1-HA was overexpressed at comparable levels in WT and NONO-KO HEK293T cells, and HOXA1 chromatin occupancy was quantified by ChIP-qPCR using an anti-HA antibody. Even in this uniform cellular context, HOXA1 binding was significantly diminished at multiple target promoters, including MESP1, PDGFRA, HAS2, and MIXL1, in NONO-KO cells (Panel A).

These results confirm that the reduction in HOXA1 ChIP signal is not solely due to cell-type composition differences but reflects an intrinsic requirement for NONO in enabling HOXA1 chromatin binding.

A ChIP-qPCR analysis showing reduced HOXA1 binding at target loci in NONO-KO HEK293T cells compared to WT. n=3. P values were calculated using two-tailed unpaired Student's t-test. $P < 0.05$ was considered significant.

Reviewer #2 - Major concerns 5:

5. The authors are recommended to validate key findings in an animal model, since the hiPSC-CM differentiation process is still quite artificial and may not truly reflect *in vivo* development.

Response:

We appreciate the reviewer's valid point regarding the importance of validating mechanistic findings in an *in vivo* context. We fully agree that hiPSC-derived cardiomyocyte systems, while powerful, do not fully recapitulate embryonic development.

However, the primary aim of this study was to delineate the molecular mechanism by which NONO cooperates with HOXA1 during early cardiac differentiation. The hiPSC model provides a highly controlled environment that allows us to isolate and interrogate these interactions without the confounding systemic effects and

compensatory pathways present in whole-animal systems. Establishing such mechanistic foundations *in vitro* is an essential step before moving to more complex *in vivo* models.

In accordance with the reviewer's suggestion, we have expanded the Discussion to explicitly acknowledge this limitation and to emphasize that testing the NONO–HOXA1 regulatory axis in an appropriate animal model represents an important direction for future research.

Discussion part:

We acknowledge that *in vitro* models cannot fully recapitulate the precise chronometry of embryonic development, nor do they possess the complex spatial architecture of native organogenesis. However, the hiPSC system offers an ideal, controlled platform for dissecting molecular mechanisms. Therefore, validating our findings *in vivo* is a critical next step. Future studies using *Nono*-deficient mice will determine whether the molecular defects observed here translate into anatomical malformations during heart development.

Corresponding descriptions have been updated in the revised text (**Page 28, lines 1088-1094**)

Reviewer #2 - Major concerns 6:

6. *The mechanism how HOXA1 and NONO are required for nuclear localization of β -catenin lacks further explanation. If the cytoplasmic β -catenin level does not change, how could it be a nuclear translocation phenotype? Alternatively, could these factors affect nuclear β -actinin stability?*

Response:

We thank the reviewer for this insightful comment. As noted, the observed reduction in nuclear β -catenin could arise either from impaired nuclear translocation or from decreased stability of the protein once in the nucleus. To distinguish between these possibilities, we examined the stability of nuclear β -catenin in the absence of HOXA1 or NONO.

We generated HOXA1-KO and NONO-KO HEK293T cell lines and performed cycloheximide (CHX) chase assays, collecting nuclear lysates at 0, 4, 8, and 12 hours after CHX treatment. These experiments revealed that loss of either HOXA1 or NONO markedly accelerates the decay of nuclear β -catenin, whereas cytoplasmic β -catenin levels remained unchanged (Panel A-D).

These findings indicate that HOXA1 and NONO are required not primarily for β -catenin nuclear import but rather for maintaining the stability of β -catenin within the

nucleus. This result provides a mechanistic explanation for the reduced nuclear β -catenin observed in NONO-KO and HOXA1-KO cells and supports the conclusion that both factors promote canonical Wnt signaling by stabilizing β -catenin in the nuclear compartment.

A, B β -catenin stability in nuclear lysates of WT and NONO-KO cells was analyzed by cycloheximide (CHX) chase assay. $n=3$. **C, D** β -catenin stability in nuclear lysates of WT and HOXA1-KO cells was analyzed by CHX chase assay. $n=3$. Data are presented as mean \pm SD. Protein levels were normalized to LAMIN A/C and expressed relative to the 0-h time point.

The results shown here in Panels A–D have been integrated into the revised manuscript as **Fig. 6E, F**. Corresponding descriptions have been updated in the revised text (**Page 23, lines 886-891**).

Reviewer #2 - Major concerns 7:

7. The authors used a classic two-small molecule differentiation methods that involve WNT activation. If NONO-KO truly impair WNT activation and thereby CM differentiation, then one would expect rescued cardiomyocyte differentiation simply by increasing the concentration of CHIR activating Wnt signaling at the first step of differentiation? This might be a stronger support for Wnt signaling mechanism.

Response:

We thank the reviewer for this insightful suggestion. To test whether enhanced WNT activation could rescue the differentiation defect in NONO-KO cells, we performed a CHIR99021 dose–response experiment during the early induction phase. Increasing the CHIR concentration from the standard condition to 10 μ M and 12 μ M produced a dose-dependent improvement in TNNT2⁺ cardiomyocyte differentiation. This effect was further confirmed by RT-qPCR, which showed significantly increased expression of TNNT2, MYH6, and MYH7 in the 10–12 μ M

groups compared with the standard condition. In contrast, increasing the CHIR concentration to 14 μM markedly reduced differentiation efficiency, consistent with previous studies showing that supraphysiological CHIR levels can induce off-target cytotoxicity and disrupt proper lineage specification¹².

Collectively, these data indicate that NONO deficiency reduces cellular sensitivity to WNT signaling, thereby raising the activation threshold required for mesoderm induction, and that this defect can be partially overcome by appropriately increasing the dosage of a WNT agonist.

A Flow cytometry analysis showing proportions of cTNT-positive cells in NONO-KO cultures treated with CHIR99021 at 8 μM , 10 μM , 12 μM , or 14 μM on day 15. **B** Quantification of cTNT-positive cells shown in (A). $n = 3$. P values were calculated using two-tailed unpaired Student's t-test. $P < 0.05$ was considered significant. **C** Expression levels of *TNNT2*, *MYH6* and *MYH7* in Day 15 NONO-KO cardiomyocytes treated with CHIR99021 at 8–14 μM , as determined by quantitative real time PCR (qRT-PCR). $n = 3$. P values were calculated using two-tailed unpaired Student's t-test. $P < 0.05$ was considered significant.

The results shown here in Panels A–C have been integrated into the revised manuscript as **Fig. 6M–O**. Corresponding descriptions have been updated in the revised text (**Page 23 and 24, lines 911–940**).

Reviewer #2 - Major concerns 8:

8. NONO genetic variants are found in CHD patients. The results in this study should be validated in a hiPSC line with patient-related genetic variants to make sure the results in this study is disease related.

Response:

We thank the reviewer for this insightful suggestion. To validate the pathogenic effects of NONO variants identified in CHD patients, and to further evaluate their functional impact during human cardiomyocyte differentiation, we performed a

DOX-inducible overexpression assay in a NONO-knockout hiPSC line. Four nonsense mutations (c.217C>T [p.Arg73*], c.550C>T [p.Arg183*], c.1009C>T [p.Arg337*], and c.1093C>T [p.Arg365*])¹³⁻¹⁵, which have been clinically associated with congenital cardiac defects including NCC, VSD, PDA, and aortic arch anomalies, were selected for functional validation. DOX-induced expression of WT NONO or the mutant variants was initiated during the early differentiation window (days 1–2).

Immunofluorescence staining revealed that NONO-KO cardiomyocytes exhibited severely disorganized sarcomeres at day 15, whereas DOX-induced WT NONO expression restored well-organized α -ACTININ/cTNT structures. In contrast, none of the four pathogenic variants were able to rescue sarcomere organization, with the majority of cells retaining disorganized sarcomere structures (Panel A). Consistent with this structural defect, flow cytometry showed a robust increase in TNNT2⁺ cardiomyocytes upon WT NONO re-expression (33.5% vs. 4.0% without DOX), whereas all four mutant variants produced only minimal TNNT2⁺ populations (8.8–11.9%) (Panel B, C). qRT-PCR revealed a robust restoration of *TNNT2*, *MYH6*, and *MYH7* expression by WT NONO, while all pathogenic variants failed to activate these sarcomeric genes (Panel D).

Taken together, these results demonstrate that NONO variants associated with CHD are functionally pathogenic and incapable of supporting cardiomyocyte lineage commitment, thereby providing direct evidence that NONO dysfunction contributes to disrupted cardiac development in humans.

A Immunofluorescence staining of α -ACTININ (green) and cTNT (red) in Day 15 hiPSC-derived cardiomyocytes generated from NONO-KO cells overexpressing WT-NONO (\pm DOX) or mutant NONO (R73*, R184*, R337*, R365*) with DOX. Quantification of cells with well-organized sarcomeres is shown on the right. Scale bar, 15 μ m. $n = 25$ (NONO-KO^{WT-NONO OE} without Dox), $n = 29$ (NONO-KO^{WT-NONO OE} with Dox), $n = 21$ (R73*), $n = 25$ (R184*), $n = 20$ (R337*) and $n = 21$ (R365*) biologically independent samples.

B Flow cytometry analysis showing increased cTNT-positive cells in NONO-KO^{WT-NONO OE} with DOX (33.5%) compared with NONO-KO^{WT-NONO OE} cells without DOX (4.0%) at day 15, while NONO mutant–overexpressing cells exhibited low cTNT-positive proportions (R73*: 8.8%; R184*: 11.2%; R337*: 11.5%; R365*: 11.9%)

C Quantification of cTNT-positive cells shown in (B). $n = 3$.

D Expression levels of TNNT2, MYH6, and MYH7 in Day 15 hiPSC-derived cardiomyocytes generated from NONO-KO^{WT-NONO OE} cells ($-$ DOX or $+$ DOX) or NONO-KO^{NONO mutants OE} (R73*, R184*, R337*, R365*) under DOX induction.

Data are presented as mean \pm s.d. (error bars). P values were calculated using two-tailed unpaired Student's t-test (C, D) or two-side Fisher's exact test (A). $P < 0.05$ was considered significant.

The results shown here in Panels A–D have been integrated into the revised manuscript as **Supplementary Fig. 5**. Corresponding descriptions have been updated in the revised text (**Page 11, lines 363-378**).

Reviewer #2 - Major concerns 9:

9. The conclusion that NONO cooperates with HOXA1 is weak. The authors

could do a reChIP experiment to see if the two molecules indeed act on the same loci. In addition, NONO-HOXA1 double silencing cells should be compared to each single KO lines to show cooperativity, rather than an upstream-downstream relationship in the same pathway.

Response:

We sincerely thank the reviewer for this constructive suggestion, which has significantly strengthened our mechanistic conclusions regarding the cooperative interaction between NONO and HOXA1.

To validate whether NONO physically co-occupies target loci with HOXA1, we performed the suggested sequential ChIP (Re-ChIP) assays using a doxycycline-inducible HOXA1-HA-overexpressing hiPSC line (WTHOXA1-HA OE) differentiated to Day 2. To ensure the preservation of protein-protein-DNA complexes during the sequential immunoprecipitation, we employed a rigorous native elution strategy: chromatin was first immunoprecipitated with an anti-HA antibody and eluted competitively using HA peptide, to maintain complex integrity. The eluate was then subjected to a second immunoprecipitation using an anti-NONO antibody. As visualized in the newly added genome browser tracks (Panel A), distinct Re-ChIP signals were successfully recovered at the regulatory regions of key cardiac development genes, including EOMES, TBXT, DVL2, and MECOM^{6,16}. Crucially, these Re-ChIP peaks precisely align with the overlapping regions observed in the individual NONO and HOXA1 ChIP-seq tracks, providing direct evidence that NONO and HOXA1 form a physical transcriptional complex on chromatin to co-regulate these key developmental genes.

A Genome browser visualization of ChIP-seq and sequential Re-ChIP tracks at key cardiac gene loci.

The results shown here in Panels A have been integrated into the revised manuscript as **Fig. 5A**. Corresponding descriptions have been updated in the revised text (**Page 19, lines 710-717**).

To examine the relationship between NONO and HOXA1, we generated a double-knockout (Double KO) hiPSC line by deleting HOXA1 exon 1 in the NONO-KO background (Panel A). qRT-PCR confirmed efficient loss of HOXA1 expression (Panel B). Upon cardiac differentiation, the Double KO cells exhibited a markedly stronger defect than the NONO-KO line alone. At Day 15, the proportion of cTNT⁺ cardiomyocytes decreased from ~12.7% in NONO-KO cultures to ~4.9% in the Double KO line (Panel C). Immunofluorescence further showed that ~96% of Double KO cells displayed disorganized sarcomeres, compared with ~75% in NONO-KO cardiomyocytes (Panel D), indicating that loss of HOXA1 accentuates the structural abnormalities caused by NONO deficiency.

A Schematic diagram of the positions and sequences of the gRNA used to generate HOXA1-KO hiPSCs. **B** qRT-PCR analysis of HOXA1 in NONO-KO and Double KO cells at differentiation day 2. $n=3$. P values were calculated using two-tailed unpaired Student's t -test. $P < 0.05$ was considered significant. **C** Flow cytometry analysis showing reduced proportions of cTNT-positive cells in NONO-KO and Double-KO at day 15. Quantification of cTNT-positive cells is shown in the right panel. $n = 3$. P values were calculated using two-tailed unpaired Student's t -test. $P < 0.05$ was considered significant. **D** Immunofluorescence staining and quantification of irregular myofibrillar structures α -ACTININ (green) and cTNT (red) in NONO-KO and Double KO hiPSCs-derived cardiomyocytes at differentiation Day-15. The percentage of cells with well-organized sarcomeres is shown in the right. Scale bar, 15 μ m. $n = 27$ (NONO-KO) and $n = 29$ (Double KO) biologically independent samples. P values were calculated by Fisher's exact test (two-sided). $P < 0.05$ was considered significant.

The results shown here in Panel A, B have been integrated into the revised manuscript as **Supplementary Fig. 1K, L**. Panel C, D have been integrated into the revised manuscript as **Fig. 5M, N**. Corresponding descriptions have been updated in the revised text (**Page 20, lines 784-789**).

Reviewer #2 - Minor Comments 1:

1. A karyotype detection of the new hiPSC lines should be supplied. It is ideal to test the results in multiple hiPSC clones, even though the authors claimed that they are similar.

Response:

We thank the reviewer for this suggestion. In response, we performed karyotype analysis on WT hiPSCs and two independent NONO-KO clones (clone 3 and clone 11).

A Karyotype analysis of WT and NONO-KO hiPSC clones (clone-3 and clone-11).

The results shown here in Panel A have been integrated into the revised manuscript as **Supplementary Fig. 1C**. Corresponding descriptions have been updated in the revised text (**Page 4, lines 109-110**).

Reviewer #2 - Minor Comments 2:

2. Fig 1F: the images do not show reduced peak intensity, but rather increased.

Response:

We acknowledge that the specific representative trace shown in Fig 1F for the NONO-KO group appears to have a robust amplitude, which visually contradicts a "reduced intensity" phenotype. When selecting this representative image, our primary intention was to highlight the most striking and statistically significant phenotypes: the drastically reduced beating frequency and the prolonged calcium decay. Notably, this specific pattern of calcium handling is consistent with previous findings; a similar trace morphology was observed in Figure 3F⁶.

Reviewer #2 - Minor Comments 3:

3. Fig2A: in the rescue experiment, the restored NONO protein level should be compared to endogenous level to determine why there is only partial rescue.

Response:

We thank reviewer for this suggestion. In response, we performed Western blot on day 2 differentiated WT, NONO-KO (clone3) and NONO-RE cells with or without DOX induction. The results show that NONO re-expression provides only a partial rescue at day 2. The results, now shown in Fig. 2A, confirm that NONO re-expression reaches only a partial level compared to endogenous NONO, consistent with the partial rescue phenotype.

A Western blot of Day-2 differentiated WT, NONO-KO (clone3) and NONO-KO^{NONO OE} without Dox induction and NONO-RE (NONO-KO^{NONO OE} with Dox induction).

The results shown here in Panel A have been integrated into the revised manuscript as **Supplementary Fig. 4B**. Corresponding descriptions have been updated in the revised text (**Page 9, lines 248-249**).

Reviewer #2 - Minor Comments 4

4. Fig3B should label clearly that the genes were all from group I. In addition, the authors should display how the rescue changes genes in each group, not just group I.

Response:

We thank the reviewer for the suggestion. We have now clearly labeled that the heatmaps show rescue-associated changes for genes in each group (A–I).

The results shown here in Panel have been integrated into the revised manuscript as **Supplementary Fig. 6A**. Corresponding descriptions have been updated in the revised text (**Page 11, lines 393-394**).

Reviewer #2 - Minor Comments 5

5. In line 569 and 618, the word “hESC” is misused. Human embryonic stem cell (hESC) is different from human induced pluripotent stem cell (hiPSC). The authors should be clear what cells are used for each cell line.

Response:

We thank the reviewer for pointing out this important correction. We apologize for the misuse of the term “hESC” and have revised the manuscript to correctly refer to the cells as “hiPSCs” in all relevant sections.

Summary and final appreciation for Reviewer #2

We are very grateful to Reviewer #2 for the careful and constructive evaluation of our work. In response to these comments, we have (i) refined our terminology to focus on cardiomyocyte differentiation rather than maturation, (ii) added time-course single-cell RNA-seq analyses to disentangle true transcriptional changes from shifts in cell composition, (iii) defined the structural basis and functional consequences of the NONO–HOXA1 interaction using domain mapping and protein stability assays, (iv) demonstrated that NONO is intrinsically required for HOXA1 chromatin binding and for maintaining nuclear β -catenin, thereby strengthening the link between the NONO–HOXA1 complex and canonical Wnt signaling, (v) tested the sensitivity of NONO-KO cells to graded Wnt activation, and (vi) functionally validated CHD-associated NONO variants in a human hiPSC-based model. We have also addressed all minor points, including karyotype analysis of edited hiPSC lines, clarification of figure labeling and grouping, and correction of cell-type nomenclature. We believe that these additions and revisions have substantially strengthened the mechanistic depth, internal consistency, and translational relevance of the manuscript. We sincerely thank Reviewer #2 again for the insightful guidance that has been instrumental in improving this study.

Reference

- 1 Lin, X. *et al.* Mesp1 controls the chromatin and enhancer landscapes essential for spatiotemporal patterning of early cardiovascular progenitors. *Nat Cell Biol* **24**, 1114-1128, doi:10.1038/s41556-022-00947-3 (2022).

- 2 Lian, X. *et al.* Robust cardiomyocyte differentiation from human pluripotent stem cells via temporal modulation of canonical Wnt signaling. *Proc Natl Acad Sci U S A* **109**, E1848-1857, doi:10.1073/pnas.1200250109 (2012).
- 3 Bedada, F. B. *et al.* Acquisition of a quantitative, stoichiometrically conserved ratiometric marker of maturation status in stem cell-derived cardiac myocytes. *Stem Cell Reports* **3**, 594-605, doi:10.1016/j.stemcr.2014.07.012 (2014).
- 4 Lescroart, F. *et al.* Early lineage restriction in temporally distinct populations of Mesp1 progenitors during mammalian heart development. *Nat Cell Biol* **16**, 829-840, doi:10.1038/ncb3024 (2014).
- 5 Lescroart, F. *et al.* Defining the earliest step of cardiovascular lineage segregation by single-cell RNA-seq. *Science* **359**, 1177-1181, doi:10.1126/science.aao4174 (2018).
- 6 Liang, Q. *et al.* Essential role of MESP1-RING1A complex in cardiac differentiation. *Dev Cell* **57**, 2533-2549.e2537, doi:10.1016/j.devcel.2022.10.009 (2022).
- 7 Delgado-Olguín, P. *et al.* Epigenetic repression of cardiac progenitor gene expression by Ezh2 is required for postnatal cardiac homeostasis. *Nat Genet* **44**, 343-347, doi:10.1038/ng.1068 (2012).
- 8 Singh, S. K., Kagalwala, M. N., Parker-Thornburg, J., Adams, H. & Majumder, S. REST maintains self-renewal and pluripotency of embryonic stem cells. *Nature* **453**, 223-227, doi:10.1038/nature06863 (2008).
- 9 Kobayashi, T. & Kageyama, R. Hes1 regulates embryonic stem cell differentiation by suppressing Notch signaling. *Genes Cells* **15**, 689-698, doi:10.1111/j.1365-2443.2010.01413.x (2010).
- 10 Qiu, X. *et al.* Single-cell mRNA quantification and differential analysis with Census. *Nat Methods* **14**, 309-315, doi:10.1038/nmeth.4150 (2017).
- 11 Gulati, G. S. *et al.* Single-cell transcriptional diversity is a hallmark of developmental potential. *Science* **367**, 405-411, doi:10.1126/science.aax0249 (2020).
- 12 Laco, F. *et al.* Unraveling the Inconsistencies of Cardiac Differentiation Efficiency Induced by the GSK3 β Inhibitor CHIR99021 in Human Pluripotent Stem Cells. *Stem Cell Reports* **10**, 1851-1866, doi:10.1016/j.stemcr.2018.03.023 (2018).
- 13 Roessler, F. *et al.* Genetic and phenotypic spectrum in the NONO-associated syndromic disorder. *Am J Med Genet A* **191**, 469-478, doi:10.1002/ajmg.a.63044 (2023).
- 14 Scott, D. A. *et al.* Congenital heart defects and left ventricular non-compaction in males with loss-of-function variants in NONO. *J Med Genet* **54**, 47-53, doi:10.1136/jmedgenet-2016-104039 (2017).
- 15 Sewani, M. *et al.* Further delineation of the phenotypic spectrum associated with hemizygous loss-of-function variants in NONO. *Am J Med Genet A* **182**, 652-658, doi:10.1002/ajmg.a.61466 (2020).

- 16 Krup, A. L. *et al.* A Mesp1-dependent developmental breakpoint in transcriptional and epigenomic specification of early cardiac precursors. *Development* **150**, doi:10.1242/dev.201229 (2023).

We sincerely thank both Reviewer #1 and Reviewer #2 for their thoughtful and constructive comments. Their insightful suggestions have significantly improved the quality of our manuscript. Below, we provide a point-by-point response to each comment. Reviewer comments are shown in *italic*.

Reviewer #1 (Remarks to the Author):

Reviewer #1 (Remarks to the Author):

In this revised manuscript, the authors have thoroughly addressed all my comments. The overall quality of the study has been subsequently improved, and the data are now clearly presented and well interpreted.

*In response to my comments, the authors have refined their conclusions regarding the role of NONO in early cardiac specification rather than late-stage maturation. Additional experiments, including new flow cytometry and single-cell RNA-seq analyses to quantify cardiac progenitor populations and cell-state transitions, have helped them better assess the role of NONO during precardiac mesoderm development. Cardiomyocytes (CM) are now better characterized using markers such as Myl2 and Myl7, which indicate that the CMs generated have predominantly atrial identity. The rescue experiments are also better characterized thanks to the inclusion of early markers. A temporal expression analysis, shown in Supplementary Figure 2, highlights the progression of CM differentiation in their model. Integration of the NONO ChIP-seq data with transcriptomic profiles has enabled the authors to identify direct targets of NONO. Clarifying the temporal relationships between NONO, MESP1, and key developmental markers has strengthened the understanding of their model and the role of NONO in CM differentiation. The discussion now includes a section on the “negative regulation of stem cell differentiation” gene set and provides a more careful comparison of their findings with *in vivo* and previous studies.*

Addressing the homodimerization of NONO and HOXA1 has further strengthened the mechanistic model of NONO–HOXA1 cooperation by characterizing their dimerization properties. Through sophisticated approaches such as Re-ChIP, they map interaction domains and assess how loss of NONO affects HOXA1 chromatin binding and downstream Wnt signaling.

Overall, the revised manuscript demonstrates significant improvements in clarity, and mechanistic insight.

We sincerely thank the reviewer for their positive assessment of our revised manuscript and for their constructive feedback throughout the review process. We are pleased to learn that the additional experiments—including the detailed characterization of cardiac progenitor populations and the mechanistic insights gained from the homodimer and heterodimer protein interaction assays—have

satisfactorily addressed your concerns. We agree that these revisions have significantly strengthened the conclusions of our study.

Reviewer #1 (Remarks on code availability):

The GitHub link provides access to the bioinformatic datasets and analysis scripts used in this study.

We thank the reviewer for confirming the accessibility of the code. we have archived the code on Zenodo as indicated in the revised manuscript. ([https://doi.org/10.5281/zenodo.18059608.](https://doi.org/10.5281/zenodo.18059608))

Reviewer #2 (Remarks to the Author):

The authors did a commendable work at addressing my comments. Most of my questions were well answered by supplemental experiments, including scRNA-seq (for the major comments 2), protein stability assays (for the major comments 3 and 6), domain-mapping experiments (for the major comments 3), mechanistic analyses (for the major comments 4), CHIR99021 dose–response experiment (for the major comments 7), validation in hiPSC lines with patient-related genetic variants (for the major comments 8), reCHIP (for the major comments 9), double knockout hiPSC lines generation (for the major comments 9) and karyotype detection (for the minor comments 1). For the major comments 5, although the authors didn't provide additional experiments, they gave a reasonable explanation in the discussion. The authors also corrected their misuse of “maturation” and “hiPSC”. The only imperfection is that replicates and a quantification are needed for Supplementary Fig. 4B. I have no further requests.

We are grateful for the reviewer's encouraging comments and are pleased that the extensive additional experiments—ranging from scRNA-seq analyses to the generation of double-knockout lines—have resolved the major concerns.

Regarding the specific request for **Supplementary Fig. 4B**, we have performed the necessary biological replicates as suggested. We have also performed the corresponding quantification analysis (~60% efficiency), which is now presented in the newly added **Supplementary Fig. 4C**.

We thank the reviewer for their thorough assessment, which has helped ensure

the robustness of our data.

A Western blot of Day 2 differentiated WT, NONO-KO (Clone3) and NONO-KO^{NONO OE} without doxycycline (Dox) induction and NONO-RE (NONO-KO^{NONO OE} with Dox induction). **B** Quantification of rescue efficiency, calculated as the ratio of NONO protein levels in NONO-RE cells relative to WT controls in panel (A). n = 4.